# MinD-RNase E interplay controls localization of polar mRNAs in *E. coli*

Shanmugapriya Kannaiah [ID][1,3,7 ✉], Omer Goldberger[1,7], Nawsad Alam [ID][1,4], Georgina Barnabas[2,5], Yair Pozniak[2], Anat Nussbaum-Shochat[1], Ora Schueler-Furman [ID][1], Tamar Geiger[2,6] & Orna Amster-Choder [ID][1 ✉]

## Abstract

The *E. coli* transcriptome at the cell's poles (polar transcriptome) is unique compared to the membrane and cytosol. Several factors have been suggested to mediate mRNA localization to the membrane, but the mechanism underlying polar localization of mRNAs remains unknown. Here, we combined a candidate system approach with proteomics to identify factors that mediate mRNAs localization to the cell poles. We identified the pole-to-pole oscillating protein MinD as an essential factor regulating polar mRNA localization, although it is not able to bind RNA directly. We demonstrate that RNase E, previously shown to interact with MinD, is required for proper localization of polar mRNAs. Using in silico modeling followed by experimental validation, the membrane-binding site in RNase E was found to mediate binding to MinD. Intriguingly, not only does MinD affect RNase E interaction with the membrane, but it also affects its mode of action and dynamics. Polar accumulation of RNase E in *ΔminCDE* cells resulted in destabilization and depletion of mRNAs from poles. Finally, we show that mislocalization of polar mRNAs may prevent polar localization of their protein products. Taken together, our findings show that the interplay between MinD and RNase E determines the composition of the polar transcriptome, thus assigning previously unknown roles for both proteins.

**Keywords** RNA Localization; Bacterial Cell Organization; Bacterial Cell Poles; RNase E; MinD
**Subject Categories** Cell Adhesion, Polarity & Cytoskeleton; Microbiology, Virology & Host Pathogen Interaction; RNA Biology

## Introduction

Research over the past few decades has shed light on the intricacy of bacterial cell organization, mainly focusing on the subcellular distribution of proteins and its implication on their activity (Govindarajan et al, 2012). However, until recently, RNAs were not assumed to have specific distribution patterns in bacteria, and the phenomenon was assumed to occur only in eukaryotes. This view changed in recent years, when tools to study localization of mRNA became available and were used to show that bacterial transcripts may display localization patterns other than near the DNA (Golding and Cox, 2004; Valencia-Burton et al, 2009; Toran et al, 2014; Irastortza-Olaziregi and Amster-Choder, 2021). Subsequent studies laid the foundation for the field of bacterial RNA localization (dos Santos et al, 2012; Montero Llopis et al, 2010; Nevo-Dinur et al, 2011; Valencia-Burton et al, 2009; Moffitt et al, 2016; Kannaiah et al, 2019). Broadly, two types of transcript localization patterns emerged from these studies: near the transcription site or to subcellular domains within the bacterial cell, each typical to specific organisms (Campos and Jacobs-Wagner, 2013). The latter class includes transcripts localizing to the membrane, cell poles and mid-cell, as well as distributed along a helical path in the cytoplasm (dos Santos et al, 2012; Kawamoto et al, 2005; Moffitt et al, 2016; Nevo-Dinur et al, 2011; Kannaiah et al, 2019; Steinberg et al, 2020; Mahbub et al, 2020; Dugar et al, 2016).

It is widely accepted that mRNA targeting to distinct locations in eukaryotic cells is crucial for various physiological processes (Buxbaum et al, 2015; Lecuyer et al, 2007; Ryder and Lerit, 2018). Intriguingly, several recent studies demonstrated that localization of bacterial mRNA often correlates with the location of their encoded proteins (dos Santos et al, 2012; Kannaiah et al, 2019; Kawamoto et al, 2005; Moffitt et al, 2016; Nevo-Dinur et al, 2011). A study by Aiba and co-workers, which showed that membrane localization of the *ptsG* mRNA is pivotal for the activity of the SgrS sRNA that efficiently destabilizes *ptsG* in response to phospho-sugar stress, provided evidence for functional implications of transcript localization in bacteria (Kawamoto et al, 2005).

In eukaryotic cells, mRNA localization is a highly regulated process mediated by a multitude of *cis*- and *trans*-acting elements (Chin and Lecuyer, 2017). Essentially, the mRNAs that carry *cis*-acting elements are recognized by *trans*-acting mRNA-binding proteins, and together they form ribonucleoprotein complexes

[1]Department of Microbiology and Molecular Genetics, IMRIC, The Hebrew University Faculty of Medicine, P.O.Box 12272, 91120 Jerusalem, Israel. [2]Department of Human Molecular Genetics and Biochemistry, Sackler School of Medicine, Tel Aviv University, 6997801 Tel-Aviv, Israel. [3]Present address: Department of Molecular Microbiology, Washington University School of Medicine, St Louis, MO 63110, USA. [4]Present address: Department of Biochemistry, University of Oxford, Oxford OX1 3QU, UK. [5]Present address: Department of Pathology, University of British Columbia, Vancouver, BC V6T 1Z4, Canada. [6]Present address: Department of Molecular Cell Biology, Weizmann Institute of Science, 76100001 Rehovot, Israel. [7]These authors contributed equally: Shanmugapriya Kannaiah, Omer Goldberger. ✉E-mail: kannaiah@wustl.edu; ornaam@ekmd.huji.ac.il

(mRNP), which are transported along the cytoskeleton to distinct locations (Buxbaum et al, 2015). However, information on RNA-localizing factors is largely missing in bacteria.

Our study of RNA localization in *E. coli* on a transcriptome-wide scale (Kannaiah et al, 2019) discovered a significant correlation between the localization of mRNA and their protein products, suggesting that RNA localization is a regulated process, and that mechanisms for transcript localization should exist also in bacteria. We further found that the polar transcriptome is unique, enriched with different RNAs when compared to the membrane and cytosolic fractions. We, therefore, decided to identify mechanisms that direct RNAs to the poles. In this study, we combined two main tactics to uncover such mechanisms in *E. coli*. One tactic was to test the localization of polar transcripts in various strains that do not express factors known to maintain cellular organization in bacteria. In parallel, we took an unbiased approach to identify proteins that co-precipitate with polar mRNAs by mass spectrometry. Together, the results from the two approaches pointed at MinD, which oscillates from pole to pole and is responsible for the correct placement of the division machinery (Ramm et al, 2019) as a factor that plays a role in the localization of polar mRNAs. We show here that MinD affects polar mRNA localization by affecting the distribution and dynamics of RNase E, the major ribonuclease in *E. coli*. Thus, in the absence of Min proteins in the cell, a significant fraction of RNase E relocates to the cell poles and accumulates there transiently, leading to selective destabilization of polar transcripts, implying that in wild-type cells the Min system favors RNase E circumferential distribution, but prevents its accumulation at the poles. MinD and RNase E have been shown to interact, but the exact site in RNase E that is involved in this interaction was not precisely defined. Rather, it was reported to lie within a 281 residues-long region (Taghbalout and Rothfield, 2007). Using a high-resolution in silico peptide–protein interaction modeling approach, we predicted a 14 residue-long RNase E segment to interact with a certain segment in MinD, and the interacting sites in both proteins were experimentally validated. Interestingly, the RNase E segment that interacts with MinD overlaps with the membrane-binding site in RNase E. In agreement with that, MinD affects the interaction of RNase E with the membrane. Moreover, MinD is shown here to affect RNase E mode of action and dynamics. Together, our findings assign novel roles to both MinD and RNase E. Finally, we show that improper localization of a polar mRNA has an impact on the cognate protein localization, raising the possibility that localized translation occurs in bacteria.

## Results

### A candidate system approach implicated a critical role for MinD and RNase E in mRNA localization

Several factors are known to play a role in bacterial cell organization, such as cytoskeletal proteins, cell division proteins, and anionic lipids that concentrate at the cell poles. Therefore, we hypothesized that one or more of these factors might be involved in the localization of polar mRNAs. To test this hypothesis, we used the MS2 system (van Gijtenbeek and Kok, 2017; Rodriguez et al, 2007) for the detection of RNAs. Briefly, the MS2 bacteriophage

protein was fused to GFP (MS2-GFP) and the mRNAs were tagged with six tandem repeats of the MS2-binding site (designated 6xbs). A transcript containing only the six repeats of the MS2-binding site, with no RNA tagged by it, served as a negative control. When co-expressed, MS2-GFP binds to the tagged RNAs and tracks its location in live cells. To study the role of these specific factors, we monitored the localization of representative polar transcripts, *bglG* and *cheA*, in strains, which do not express the factor or express a non-functional factor, by live cell microscopy.

Cytoskeletal and its associated motor proteins, dynein, kinesin, and myosin, were shown to be critical for active transport of mRNA in higher organisms (Jansen and Niessing, 2012; Suter, 2018). *E. coli* codes for MreB, an actin structural homolog that maintains the rod morphology of the cell (Wachi et al, 1989). To study the putative role of MreB in polar RNA localization, we monitored the localization of the representative polar transcripts in cells treated with A22, which disrupts MreB polymerization and leads to loss of the rod shape (Bean et al, 2009). The results in Figure EV1A show that the polar transcripts still formed foci in the A22-treated spherical cells. To see if MreB co-localizes with the mRNA foci in these cells, we monitored the localization of the mRNAs in a strain that encodes MreB-RFP$^{SW}$ [a sandwich fusion protein (Morgenstein et al, 2015)] before and after treatment with A22, i.e., in rod and in spherical cells. Our results show that the tested mRNAs and MreB did not colocalize (Figure EV1A). Hence, MreB does not seem to play a role in polar mRNA localization.

To test the putative role of the RNA chaperone Hfq in polar mRNA localization, we monitored the localization of the representative polar transcripts in Δ*hfq* cells. The results in Figure EV1B show that the absence of Hfq did not affect the localization of the polar mRNAs that we studied, ruling out its involvement in their localization.

Cardiolipin is an acidic phospholipid that localizes to *E. coli* membrane regions of high negative curvature, found mainly at the poles (Renner and Weibel, 2011). To check if cardiolipin acts as a cue for the localization of polar transcripts, we used a strain that is deficient in its synthesis and found that the tested mRNAs remained mainly at the poles (Figure EV1C), excluding its role in mRNA localization.

To test for the putative role of the Min system, which is responsible for the correct placement of the division machinery, we monitored the localization of the representative polar transcripts in a strain deleted for *minCDE*. This strain produces filamented cells, because cell division is impaired in these cells (Raskin and de Boer, 1999). The results in Fig. 1 show that the localization of both *bglG* and *cheA* mRNA was affected in the Δ*minCDE* cells. Thus, unlike the polar localization observed in wild-type cells (Fig. 1A), they displayed diffuse distribution throughout the cell, similar to the no-tagged mRNA control (Fig. 1B).

Because the Min system is primarily known for its involvement in cell division, we asked whether cell division per se is involved in the localization of polar transcripts. To address this question, the cells were treated with cephalexin that inhibits FtsI, a cell division protein that synthesizes peptidoglycan at the division site, whose inactivation inhibits the constriction of the Z-ring and causes filamentation. The results in Fig. 1D show that the transcripts maintained their polar localization in cephalexin-treated cells, indicating that cell division per se does not regulate polar transcript localization.

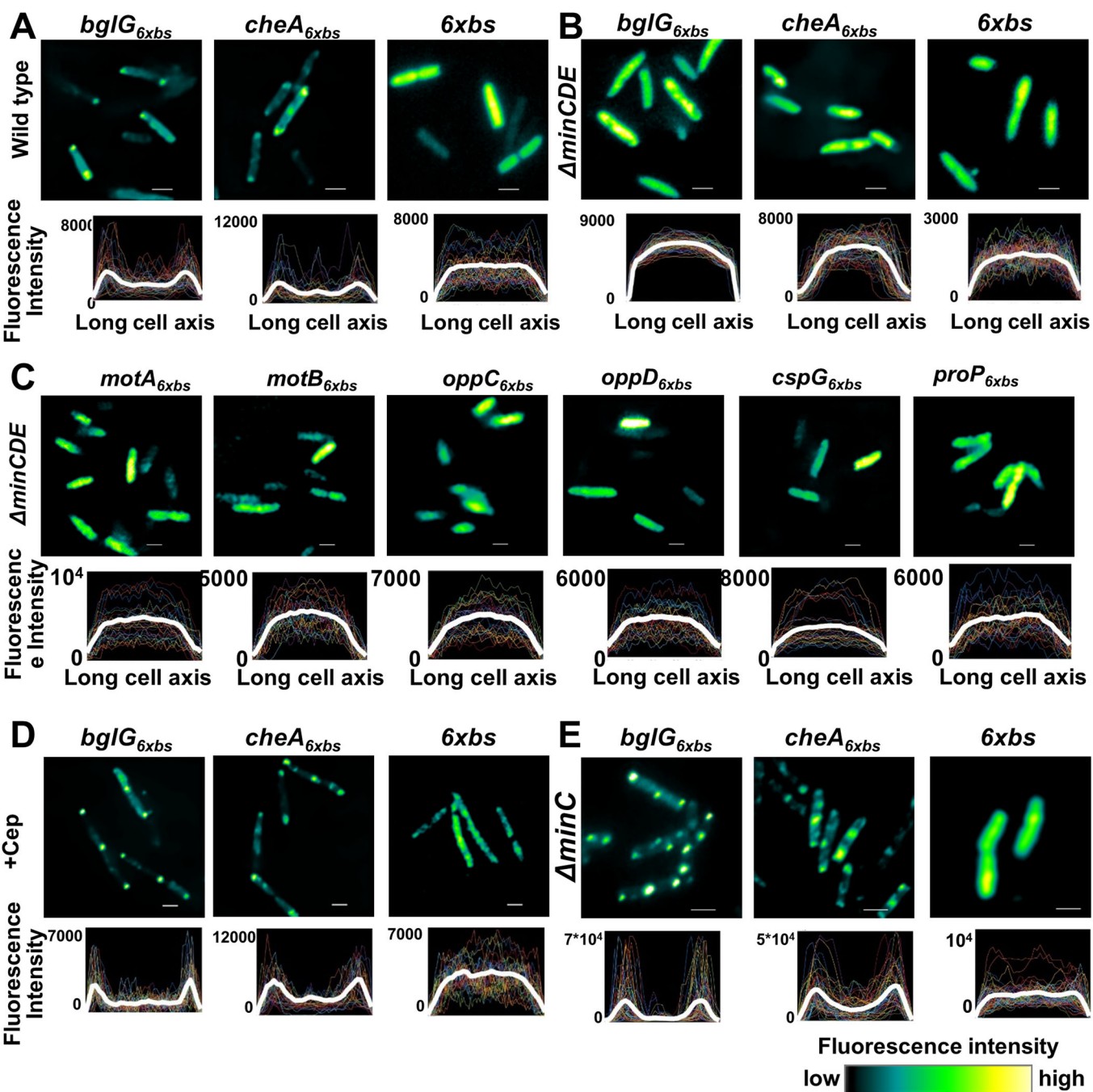

**Figure 1. Polar transcripts are diffused in Δ*minCDE* cells, but not in Δ*minC* cells.**

(A) Upper panel: Images showing localization of representative polar transcripts, *bglG* and *cheA*, as well as no mRNA control (six binding sites for the MS2 protein, designated 6xbs) in wild-type cells. Lower panel: The average fluorescence intensity profiles of the transcripts plotted against the long cell axis after normalizing to cell length; $n = 90$–100. (B, C) Upper panels: Images showing localization of transcripts in Δ*minCDE* cells: *bglG* and *cheA*, as well as no mRNA control (6xbs) (B); *motA*, *motB*, *oppC*, *oppD*, *cspG* and *proP* (C). Lower panels: The average fluorescence intensity profiles of the transcripts plotted against the long cell axis; $n = 90$–100. (D) Upper panel: Images showing localization of representative polar transcripts, *bglG*, *cheA*, as well as no mRNA control (*6xbs*) in cells treated with cephalexin (Cep). Lower panel: The average fluorescence intensity profiles of the transcripts plotted against the long cell axis after normalizing to cell length; $n = 50$–60. (E) Upper panel: Images showing localization of representative polar transcripts, *bglG*, *cheA*, as well as no mRNA control (*6xbs*) in Δ*minC* cells. Lower panel: The average fluorescence intensity profiles of the transcripts plotted against the long cell axis after normalizing to cell length; $n = 90$–100. mRNAs were detected in live cells by the MS2 system. Images are representatives of biological triplicates (A–E). Scale bar corresponds to 2 μm. The fluorescent signal is presented as shown in the color key in the fluorescence intensity heat map. Colored lines in the average fluorescence intensity profiles represent the fluorescence intensity of individual cells, whereas the thick white line represents the averaged fluorescence intensity against the long cell axis. Source data are available online for this figure.

To distinguish the effect of MinD from that of the other Min proteins, we monitored the localization of our test transcripts, *bglG* and *cheA*, in Δ*minC* cells. Knockout of the *minC* gene had no visible effect on the polar localization of these transcripts (Fig. 1E), ruling out its role in polar mRNAs localization. Importantly, the deletion of *minC* does not have a polar effect on the expression of *minD* (Figure EV2). The *minE* gene cannot be deleted, because the production of MinC and MinD in the absence of MinE will prevent Z ring formation all along the membrane causing lethal cell filamentation (de Boer et al, 1989). To substantiate the role of the Min system in the localization of polar mRNAs, we monitored the localization of several additional polar RNAs in Δ*minCDE* cells (Fig. 1C). Polar localization of all the transcripts that were tested was completely abolished in these cells. Together, these results substantiate the role of the Min system in localizing polar transcripts.

Finally, we examined the involvement of the RNA degradosome in the localization of polar transcripts. The RNA degradosome in *E. coli* is composed of the major RNA ribonuclease, RNase E, and three auxiliary proteins, enolase, RhlB and Pnp, which together are involved in RNA processing and decay (Carpousis, 2007). RNase E has been observed to localize near the cell membrane in foci, which are assumed to be RNA-protein complexes (Khemici et al, 2008). It was also shown in *Caulobacter crescentus* that the cytoplasmatic RNase E forms bacterial RNP (BR) bodies composed of the RNA degradosome constituents and RNA (Al-Husini et al, 2018). Because RNase E is essential, we monitored the localization of polar transcripts in a temperature-sensitive (ts) mutant of RNase E (*rne3071*). The results in Fig. 2A show that at the restrictive temperature (42 °C), in which RNase E is non-functional, the polar transcripts became diffused, as opposed to their polar localization at the non-restrictive temperature (30 °C). Shifting to 43 °C has been previously used to inactivate RNase E in this mutant (McDowall et al, 1993). We validated that it is also inactivated at 42 °C by showing that the 9 S rRNA precursor accumulates because it is not cleaved to generate p5S transcript both at 42 °C and 43 °C (Cormack and Mackie, 1992; Appendix Fig. S1). Notably, mutants impaired or deleted for the other degradosome components, *eno-2* cells that encode a non-functional enolase protein and Δ*pnp* and Δ*rhlB* cells, exhibited the same phenotype as *rne3071* cells, that is, the mRNAs were diffused rather than at poles (Figs. 2B,C and EV3). Since the polar transcripts became diffused in all strains lacking or expressing non-functional degradosome components, we decided to check if the polar foci of the transcripts in wild-type cells co-localize with the degradosome clusters at the poles. To this end, we monitored the localization of RNase E and of the polar transcripts simultaneously. The results in Fig. 2D show that the polar transcripts do not colocalize with the clusters of RNase E at the cell poles, indicating that the polar RNAs are not part of the degradosome complexes. Taken together, our approach points out that the Min system and the degradosome complex are involved in mRNA localization.

## A high-throughput proteomic approach detected MinD as associated with polar transcripts

In parallel to the biased candidate system approach, we took an unbiased proteomics-based approach to identify proteins involved in the localization of polar transcripts. For this approach, we used the constructs used for visualization of the polar transcripts in live cells, which are tagged by repeats of the sequence recognized by the RNA-binding MS2 protein fused to GFP (MS2-GFP). The constructs were transformed into the χ1488 strain, which divides asymmetrically near the cell poles at a high frequency, resulting in the production of minicells that encapsulate the content of the cell poles (Meagher et al, 1977). We have previously demonstrated that the transcripts that are enriched in the minicells formed by χ1488 cells are indeed pole-enriched RNAs (Kannaiah, 2019). After isolating the minicells, proteins associated with the polar transcripts were pulled down, using antibodies against GFP, and the proteins that co-precipitated with the MS2-GFP were purified.

To verify that this procedure captures the polar mRNAs, we extracted RNA from the immunoprecipitation eluate of minicells, which were isolated from cells harboring plasmids that express certain mRNAs tagged by MS2-binding sites, only the MS2-binding sites not tagged to any mRNA (6xbs) or the same mRNAs not tagged by MS2-binding sites (the latter two served as controls). The extracted RNA preparations were then subjected to reverse transcription followed by PCR (RT-PCR), using primers that amplify these mRNAs. The results in Appendix Fig. S2 demonstrate that *bglG* mRNA was observed only in the eluates from cells expressing this mRNA tagged with MS2-binding sites, thus ensuring that the procedure allows for the isolation of specific mRNAs and the proteins bound to them.

After having validated the specificity of the procedure, the proteins that co-precipitated with the polar mRNAs were identified by mass spectrometry, as depicted in the schematic representation in Fig. 3A. For the *bglG* mRNA, 42 proteins were detected with high significance, including BglG (Fig. 3B). Notably, MinD was identified as one of the statistically significant proteins that co-precipitated with the *bglG* transcript. Expectedly, many proteins associated with mRNA lifecycle were detected (Fig. 3C). In light of the activity of BglG as a transcriptional antiterminator (Amster-Choder, 2005), the pulldown of two RNA polymerase subunits provides reassurance to the results. For the *cheA* transcript, less proteins were detected with high significance. The results in Figure EV4 present the list of enriched proteins purified with *cheA* (Figure EV4A), as well as GO terms that are enriched (Figure EV4B). MinD was identified in the *cheA* samples, but did not pass the statistical significance threshold that we set (*p* value > 0.01). The two lists, obtained for *bglG* and *cheA*, share several proteins, among them two 50S ribosomal proteins (RpmG, RpmI) and three chemotaxis-related proteins (Tap, Trg, CheA), all known to be enriched at the poles.

## RNase E transiently accumulates at the poles of Δ*minCDE* cells

MinD was suggested to bind to DNA (Di Ventura et al, 2013). To check if MinD can also bind RNA, we immunoprecipitated MinD-GFP using GFP-Trap beads and checked for the presence of our representative polar transcripts, *bglG* and *cheA*, by qPCR (Tree et al, 2014). We did not detect these transcripts in the MinD pulldown, thus ruling out the possibility that MinD is directly involved in targeting them to the cell poles. Therefore, we hypothesized that MinD is involved in polar localization of transcripts by affecting another factor or a process.

It has been reported, based on yeast two-hybrid results, that MinD binds to RNase E (Taghbalout and Rothfield, 2007).

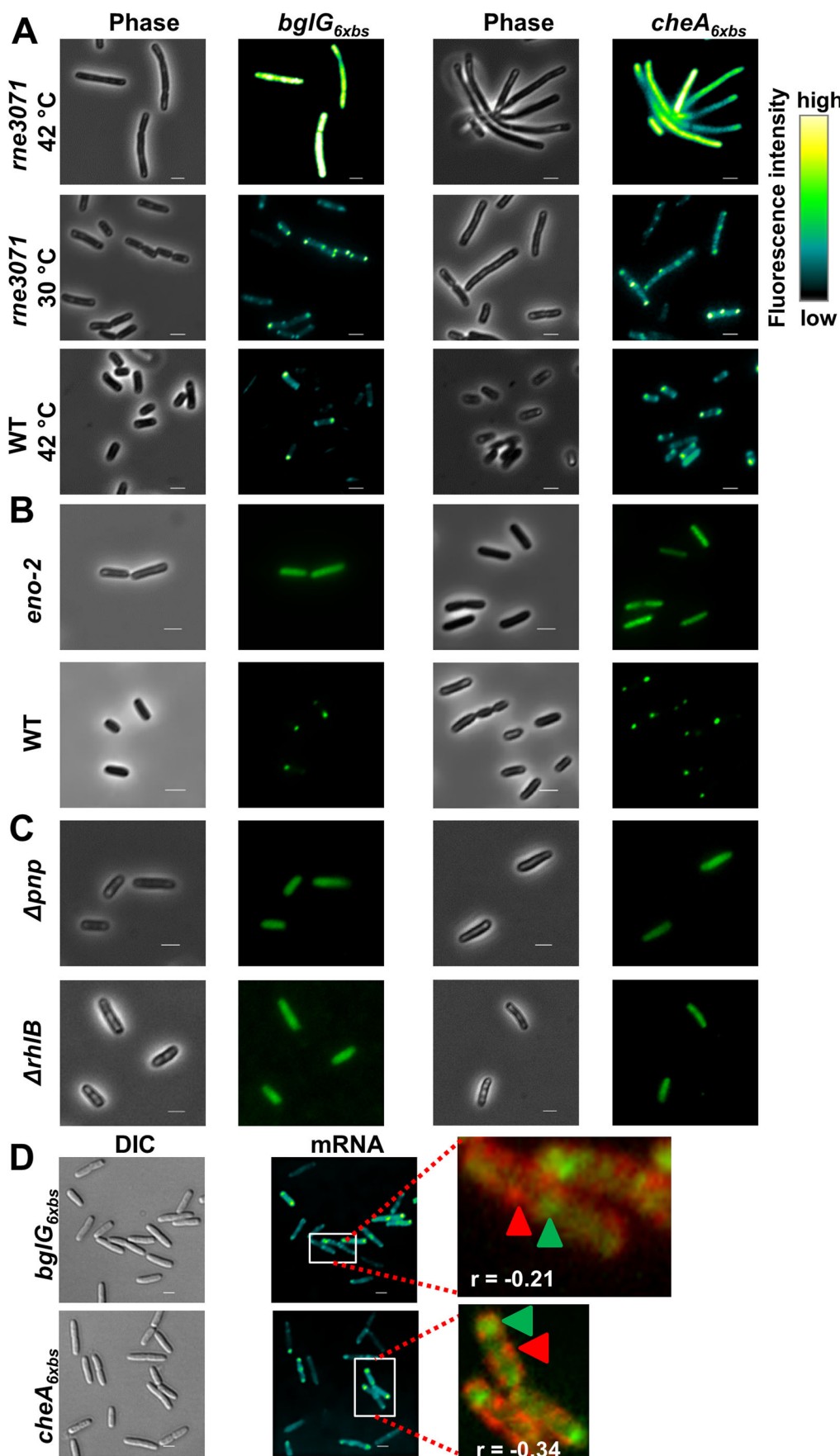

**Figure 2.  Spatial distribution of polar transcripts is affected by deletion or disruption of the functional activity of the degradosome components, but they are not part of the degradosome complex.**

(A–C) Images showing localization of representative polar transcripts, detected by the MS2 system, in live cells: (A) *rne3071* mutant and its parental strain at restrictive (42 °C) and permissive (30 °C) temperature; (B) *eno-2* mutant and its parental strain (WT); and (C) Δ*rhlB* and Δ*pnp* deletion mutants. (D) Images showing representative polar transcripts, detected by the MS2 system (green), in live cells expressing RNAse E-mCherry (red). Overlays of RNase E (red) and mRNA (GFP) in the magnified images on the right show no overlap between the two. Cells are shown as DIC images (gray). Pearson's correlation coefficient (designated by *r*) calculated by NIS-elements presented for each strain. Images are representatives of biological triplicates. Scale bar corresponds to 2 µm. Source data are available online for this figure.

Therefore, we decided to test whether MinD affects RNase E subcellular distribution. Indeed, in Δ*minCDE* cells, we observed that RNase E transiently accumulated at the cell poles, in addition to its presence near the membrane all around the cell circumference. This distribution differs from wild-type cells, in which RNase E predominantly localizes near the membrane, with the poles not being enriched, and in fact, quite depleted for RNase E compared to non-polar membrane regions (Fig. 4A, Movie EV1). To verify the transient nature of RNase E polar clusters, we adopted a previously used approach to study MreB and RNase E movement, which is to use the movie for the generation of a kymogram (Strahl et al, 2015b; van Teeffelen et al, 2011; Fig. 4B). Of note, as opposed to the relocation of RNase E to the poles in Δ*minCDE* cells, overexpression of MinD did not show an effect on RNase E spatial distribution (Appendix Fig. S3).

To check if the polar accumulation of RNase E is due to changes in its expression in cells lacking the Min system, we monitored RNase E levels in wild type and Δ*minCDE* in the different growth phases. No significant change was observed in the RNase E level between the wild type and Δ*minCDE* cells under any tested condition (Appendix Fig. S3). Hence, the polar accumulation of RNase E in Δ*minCDE* cells is not due to changes in its expression level, but rather due to its relocation to the poles in the absence of the Min system, suggesting that the Min proteins affect RNase E distribution.

The accumulation of RNase E at the poles of Δ*minCDE* cells may explain the diffused distribution of otherwise polar mRNAs in these cells (Fig. 1B,C). All these mRNAs seem to distribute in the cytoplasm of Δ*minCDE* cells, with the lowest signal at the poles, where RNase E peaks, suggesting that their polar localization in wild-type cells is due to a lower level of RNase E in this domain. To test this hypothesis, we calculated the half-lives of various transcripts in wild-type and Δ*minCDE* cells. The results in Fig. 4C show that transcripts that have been previously shown by us to be enriched at the poles (Kannaiah et al, 2019) (*cheA*, *gadE*, and *motA*) are selectively destabilized in Δ*minCDE* cells. On the contrary, transcripts that are not enriched at the poles (*adhE*, *gyrA*, *fabB*) do not present this phenomenon, supporting the possibility that, by preventing RNase E accumulation and activity at the cell poles, MinD affects not only mRNAs polar localization, but also their cellular level.

### A short peptide within the membrane-targeting sequence of RNase E mediates binding to MinD

The region of RNase E that was previously identified to interact with MinD is considerably large. It is comprised of amino acids 378–659, spanning a portion of the N-terminal catalytic domain, the entire membrane targeting sequence (MTS) (termed segment A) and a portion of the RNA-binding domain of RNase E (Taghbalout and Rothfield, 2007). To predict the site of interaction

between the two proteins, the crystal structure of MinD complexed with a MinE peptide (residues 12–31) [PDB id: 3r9i, Park et al, 2011] was used as the template. Given the highly unstructured nature of the RNase E region of interest (see Appendix Fig. 4), we assumed that the nature of MinD-binding site and the mode of RNase E binding to MinD are similar to the nature and mode that characterize the binding of MinE to MinD, e.g., that it can be simulated as an interaction between the MinD protein and a peptide within RNase E. Hence, we used a high-resolution in silico peptide-protein interaction modeling algorithm, termed FlexPep-Bind (Alam et al, 2017) to predict the site in RNase E that interact with MinD (Fig. 5A). The peptide that showed the strongest binding propensity to MinD, which extends from residue 568 to residue 582 in RNase E, overlaps with segment A (residues 565–585), as shown in Fig. 5B. The implication of these in silico results is that the same site in RNase E that binds to the membrane is also employed for binding MinD.

Of note, we could not introduce a deletion of *minD* in a strain that is deleted for RNase E and expresses an RNase E that lacks segment A from a low copy plasmid (Δ*rne*/P~rne~-RNase EΔA) (Khemici et al, 2008), nor could we overexpress MinD exogenously in the Δ*rne*/P~rne~-RNase EΔA stain, which expresses the *minD* gene from the chromosome. In both cases, the engineered strains did not survive, at least under the growth conditions that we applied.

To experimentally examine the *in-silico* prediction results, that is, to ask if MinD interacts with segment A in RNase E, we took two approaches. The first was to investigate the capacity of MinD and RNase E to interact directly and the role of segment A in RNase E in this interaction by conducting a Far-Western analysis. To this end, lysates of cells expressing either RNE', a truncated RNase E protein (residues 378–659), which contains segment A and was shown to interact with MinD (Taghbalout and Rothfield, 2007), or an RNE' deleted for segment A (RNE'ΔA) were subjected to SDS-PAGE and blotted onto a membrane, which was incubated with purified FLAG-tagged MinD and antibodies against the FLAG tag (Appendix Fig. S5). The results in Fig. 5C and Appendix Fig. S6 show that MinD interaction with RNase E is abolished when segment A is deleted, highlighting the importance of this region in the direct interaction between MinD and RNase E. Of note, RNE'ΔA has been shown before to be stably expressed and maintained at a slightly higher level than the wild type variant (Khemici et al, 2008). As a second approach, we tested the interaction by a bacterial two-hybrid assay (Battesti and Bouveret, 2012), using one plasmid that expresses MinD, and a second plasmid that expresses either RNE' or RNE'ΔA. The results in Fig. 5D show that whereas MinD interacted with RNE', as manifested by the growth of blue colonies on X-Gal plates due to expression of the β-galactosidase reporter gene, MinD failed to interact with RNE'ΔA, implying that the interaction of MinD with RNase E in vivo depends on segment A.

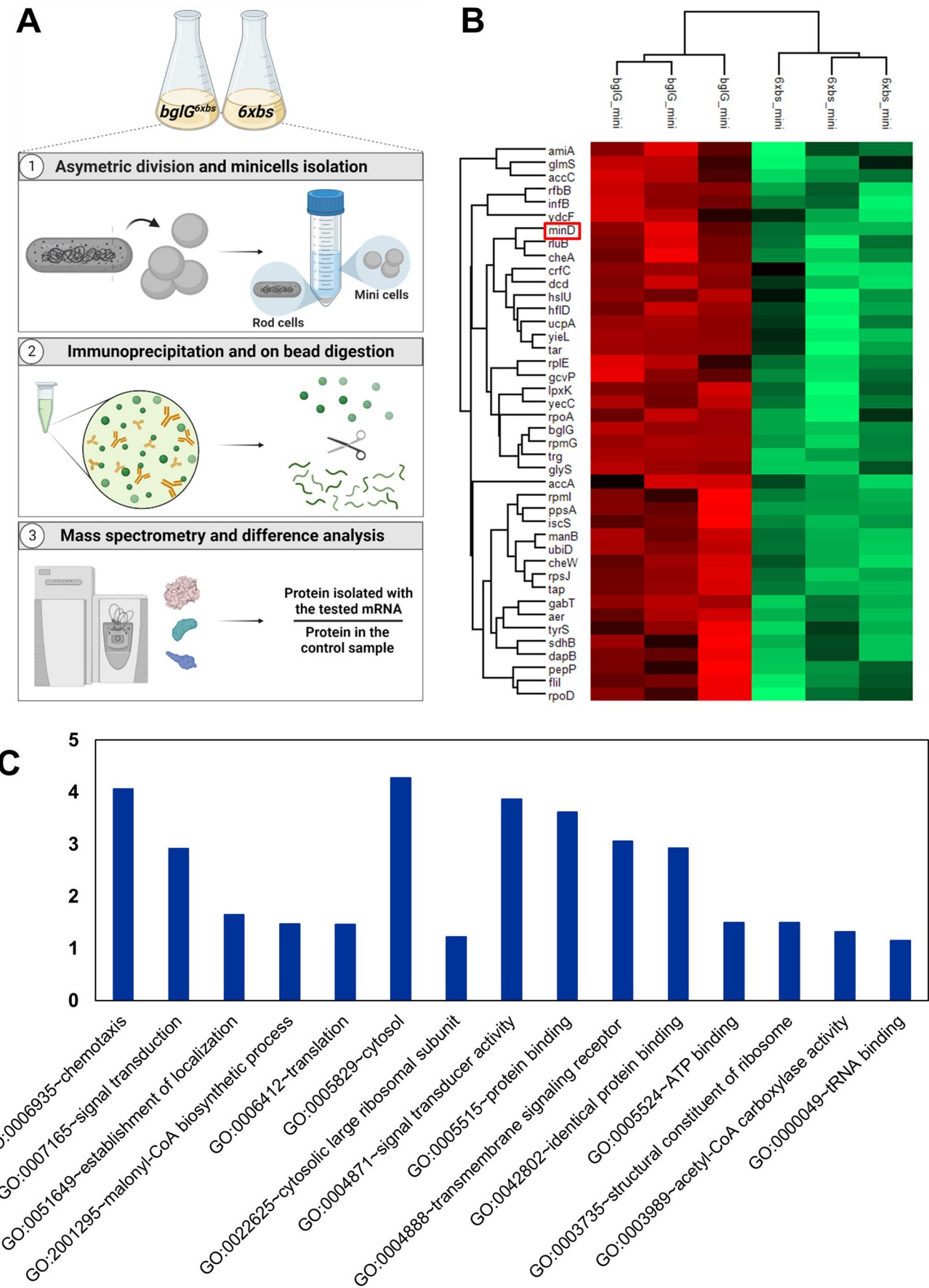

**Figure 3.  High-throughput proteomics screen to identify RNA-binding proteins.**

(A) Schematic representation depicting the procedure applied to immunoprecipitate RNA-bound proteins and their detection. For each protein, the ratio between the fraction purified with the tested mRNA over the fraction in the control sample was calculated. (B) Heat map showing differentially enriched proteins that were pulled down with the *bglG* mRNA and with the no mRNA control (6xbs); *p* value < 0.01. The MinD protein is highlighted by a red frame. (C) Graphs showing statistically significant GO terms for the immunoprecipitated proteins that were identified by mass spectrometry as binding to the *blgG* RNA in minicells.

To further confirm our predicted mode of interaction between the MinD protein and RNase E peptide, three amino acids in the predicted binding pocket of MinD: L48, V147 and L194 (Fig. 5A), which were also shown to be involved in MinC activation and in response to MinE (Wu et al, 2011), were selected for mutagenesis. We used a bacterial two-hybrid assay with constructs expressing MinD with these amino acids substituted to arginine. The results in Fig. 5E show that the replacement of V147 and L194 in MinD, which are predicted to directly interact with residues in RNase E (Fig. 5A), with arginine, abolished the interaction with RNE' when compared to wild-type MinD. Replacement of L48 in MinD, predicted to be at the interaction site with MinD, but not to interact with a specific residue in RNase E (Fig. 5A), reduced the interaction, but did not abolish it completely. The level of the MinD variant proteins in the cell was compared to that of the wild type to rule out the possibility that lack or lower expression and/or reduced stability of the mutants account for the differences in the interaction with RNE' containing or not containing segment A. The results in Appendix Fig. S7 show that the level of the wild type and mutants are comparable. Notably, the point mutations in MinD also do not cause alteration in RNase E distribution in the cell (Fig. 5F). These results corroborate our prediction regarding the MinD site that interacts with RNase E and, together with the other results described above, validate our prediction of the interaction between the two proteins.

## MinD affects the interaction of RNase E with the membrane and its mode of action

Our in silico prediction, delimiting the MinD-binding site to a peptide within the MTS of RNase E, suggests that its interaction with MinD and with the membrane are either related or mutually exclusive. The weak contacts that RNase E forms with the lipids in the inner membrane, which enable its free and rapid diffusion in the membrane (Strahl et al, 2015a) on one hand, and its apparent weak interaction with MinD, which enables independent organization of the two proteins (Taghbalout and Rothfield, 2007) on the other hand, can support either option. To examine the possibility that MinD affects the presence of RNase E in the membrane, we checked whether the extent to which RNase E is present in the membrane fraction of ΔminCDE cells is lower than this fraction in wild-type cells by comparing the RNase E fluorescent intensity in the two strains. The results in Fig. 6A show that the level of RNase E in the membrane fraction of ΔminCDE cells is 30% lower than its level in wild-type cells. To validate the reduction in the fraction of RNase E in the membrane of ΔminCDE cells, we fractionated wild-type and ΔminCDE cells to cytoplasm and membrane fractions and determined RNase E levels in two fractions, as well as the total level in the cell before fractionation, by Western blot analysis. The result in Fig. 6B shows that, whereas the total level of RNase E in the two strains is comparable, there is a decrease in RNase E in the membrane fraction of ΔminCDE cells and a reciprocal increase in

the cytoplasmic fraction compared to wild-type cells. These results suggest that MinD is involved in the efficient targeting of RNase E to the membrane. Estimates regarding the copy numbers of RNase E and MinD, derived from genome-wide ribosome profiling, which indicate that the number of RNase E molecules is approximately twofold lower than that of MinD [(Li et al, 2014) and https://biocyc.org/group?id=biocyc17-463-3725652132], provide some rational for the extent of MinD influence over RNase E.

To see if the Min system affects RNase E dynamics when associated with the membrane, we performed a FRAP (fluorescence recovery after photobleaching) experiment. Following photobleaching of RNase E foci in membrane regions, both polar and non-polar, the RNase E-YFP fluorescence foci recovered in the membrane of the wild-type cells, but no recovery of RNase E-YFP fluorescence was observed in ΔminCDE cells for the entire duration of the experiment (8 min) (Fig. 6C). These results suggest that the Min system is involved, directly or indirectly, in RNase E dynamics/diffusion in the membrane, although the lack of recovery in ΔminCDE cells might also be related to the effect of the Min system on the association of RNase E with the membrane.

Notably, MinD pole-to-pole oscillation and RNAse E dynamics in the membrane are not correlated, as can be seen in Movie EV2, when MinD-GFP oscillation and RNase E-mCherry movement in the membrane were monitored simultaneously. In agreement with that, RNase E puncta distribution along the membrane in cells expressing a non-oscillating ATPase-deficient MinD mutant (MinD$^{K16Q}$) (de Boer et al, 1991) is comparable to RNase E distribution in wild-type cells (Fig. 6D).

During normal cell growth, RNase E forms visible puncta along the inner-cytoplasmic membrane, which are apparently degradosome bodies comprising of RNA-protein complexes (Strahl et al, 2015a). It was reported that when transcription is inhibited, the RNase E clusters in the membrane disappear and RNase E spreads uniformly in the membrane all along the cell perimeter, due to the resultant depletion in RNA substrates for degradation and the presence of a bigger fraction of RNA-free RNase E (Strahl et al, 2015a). To check if deletion of the Min proteins would have an effect on the capability of RNase E to release RNAs, we inhibited transcription by the addition of rifampicin and monitored RNase E localization in ΔminCDE. Surprisingly, RNase E was observed in clusters along the membrane in ΔminCDE cells after inhibiting transcription (Fig. 6E, left), as opposed to the smooth diffusion observed in wild-type cells (Fig. 6E, right and Strahl et al, 2015a). These results suggest that MinD is required for proper release/degradation of RNAs from/by RNase E, that is, MinD might affect RNase E-mediated turnover of transcripts. This is apparently relevant mainly or only for polar transcripts, since in wild-type cells, as opposed to ΔminCDE cells, RNase E is almost depleted from the poles. Alternatively, MinD might facilitate the diffusion of RNA-free RNase E in the membrane. The latter possibility seems less likely, because it is hard to come up with an explanation for why MinD would affect the diffusion of RNA-free RNase E and not

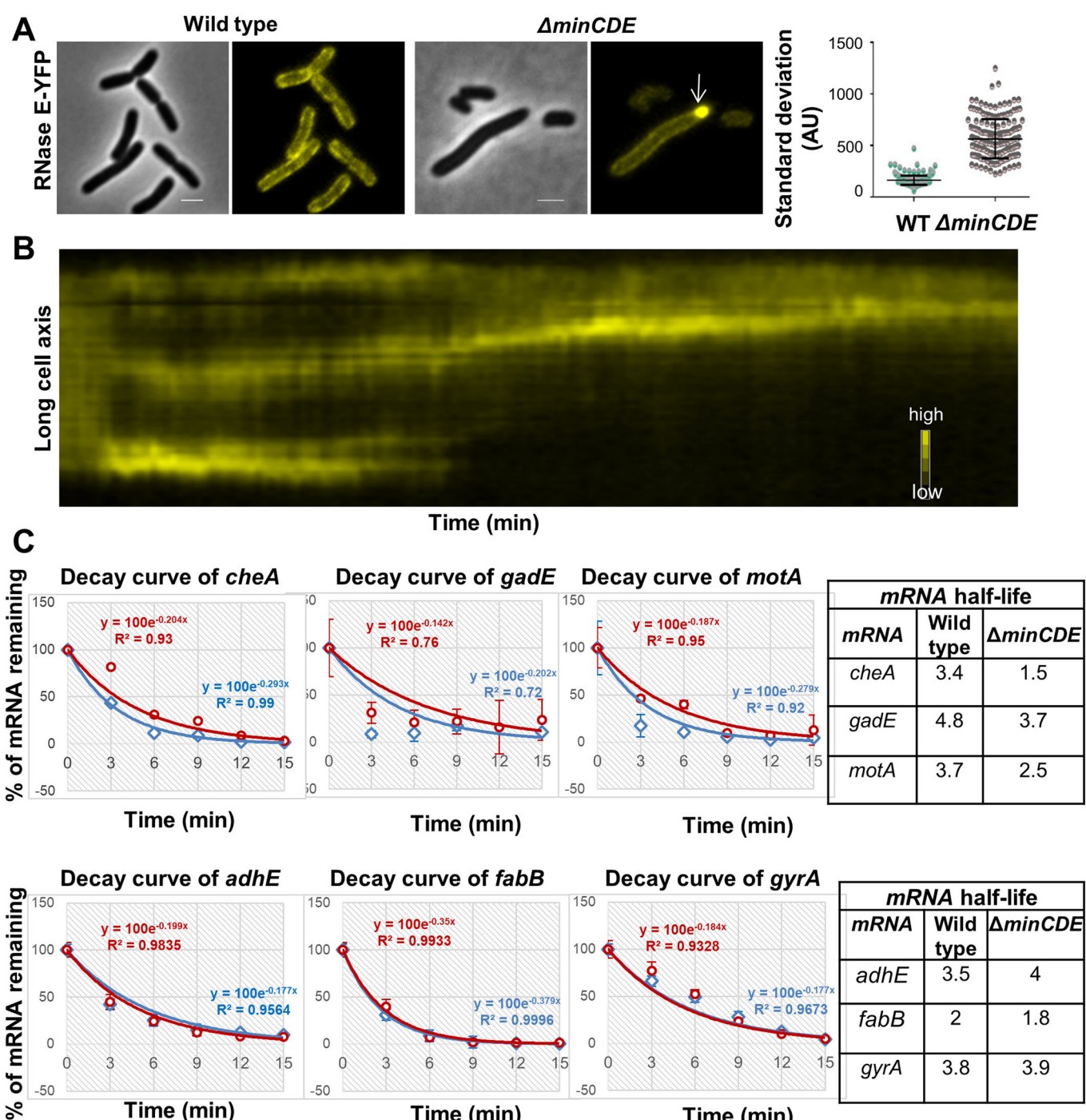

**Figure 4.  RNase E transiently localizes to Δ*minCDE* cell poles and affect polar *mRNA* stability.**

(**A**) Left panel: Membrane localization of RNase E-YFP in wild-type cells. *Center panel*: Polar foci (white arrow) formed by RNase E in Δ*minCDE* cells. *Right panel*: Standard deviation (*Y* axis, SD) was calculated for the fluorescence intensity inside the cell using NIS-elements; the standard deviation of fluorescence intensity of a cell is indirectly proportional to its uniform distribution. The increase in SD reflects the increase in polar cluster formation in Δ*minCDE* cells compared to wild-type (WT) cells. 12% of the cells present a polar cluster, *n* > 540. Images are representatives of biological triplicates. Scale bar corresponds to 2 μm. The horizontal line in each plot indicates the mean with standard deviation range. Statistical analysis for the differences between cell populations was performed using Mann−Whitney test. The calculated *p* value is <0.001 for both strains. (**B**) Single-cell kymograph of RNase E-YFP based on Movie EV1. The kymograph presents the fluorescent intensity along the cell long axis (*Y* axis) over time (*X* axis, 0–300 min). (**C**) Decay of polar (upper row) and non-polar (lower row) mRNA levels over time in wild-type (WT) and Δ*minCDE* cells as examined by qPCR. Half-lives were calculated based on smoothed curve fits (see Methods section) and are presented in the tables on the right. Source data are available online for this figure.

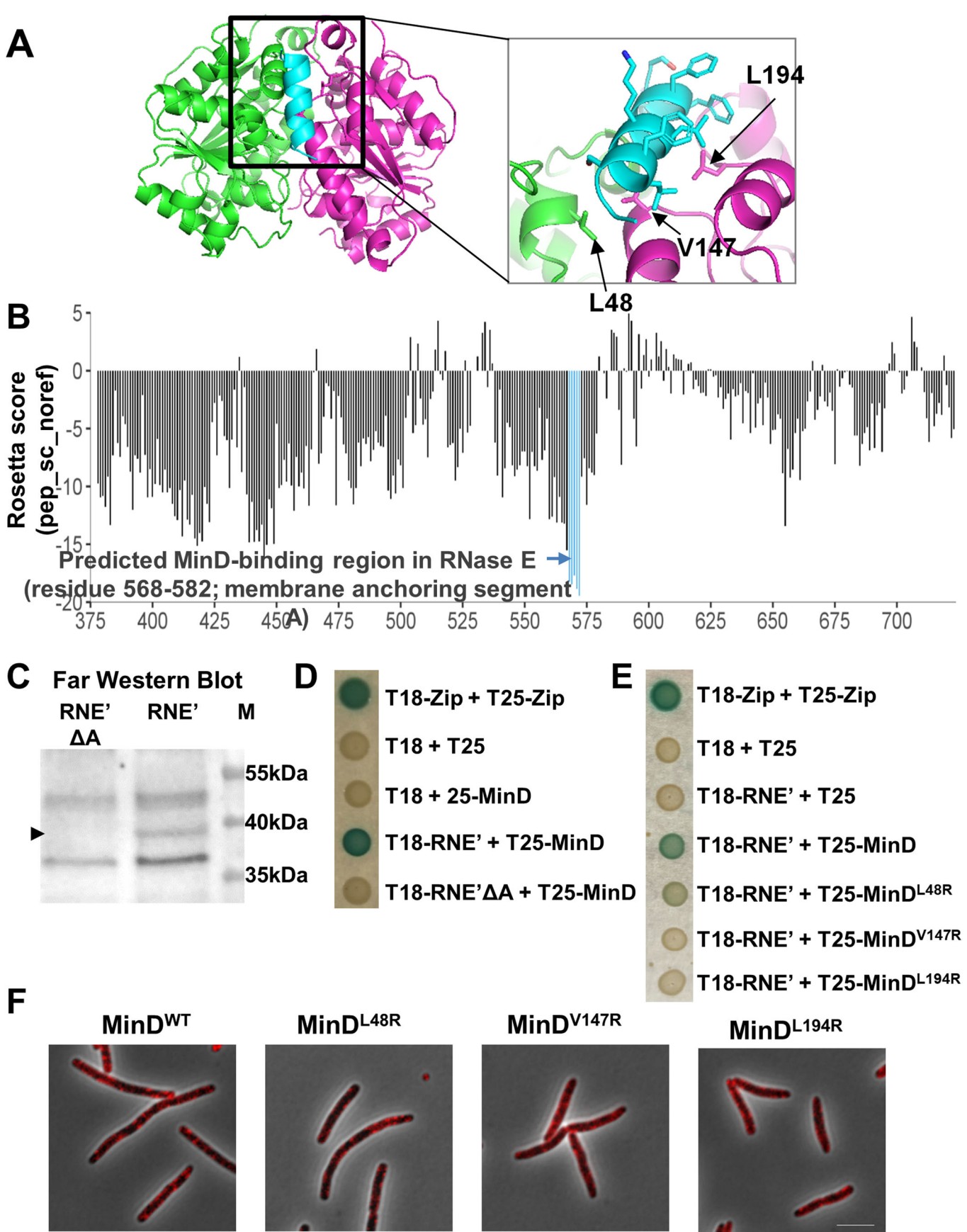

**Figure 5.   MinD interaction with RNase E is mediated by segment A of RNase E.**

(A) Predicted model for RNase E-MinD interaction: individual chains of the monomers that compose the MinD dimer are colored as green and magenta, whereas the RNase E segment 572-585 is shown in cyan; the MinD residues mutated to arginine are indicated with arrows. (B) Binding energies of 14-mer RNase E peptides (sliding windows through residues 378–724) to MinD, calculated using the FlexPepBind protocol. The best scoring windows (pep_sc_noref ≤−17.5; shown in blue) encompass residues 568-582, clustered within the RNaseE membrane anchoring region (segment A). (C) Far-Western analysis of the interaction between MinD and RNE' or RNE'ΔA. Lysates of cells overexpressing His-tagged RNE' ΔA (lane 1) or RNE' (lane 2) from the lac promoter in pET15b were separated by SDS-PAGE, blotted onto a nitrocellulose membrane and probed with MinD-FLAG and then with an anti-FLAG antibody. The band that matches His-RNE' size (~40 kDa) is marked by an arrowhead. M—protein markers (lane 3). (D) Bacterial two-hybrid assay of RNE'-MinD protein-protein interaction. BTH101 strain co-expressing RNE' or RNE'ΔA with MinD from plasmids fused to adenylate cyclase fragments T18 and T25, respectively, as well as a leucin zipper (Zip) domain fused to T18 and T25 (positive control) and the two empty vectors (negative control) were used for the interaction studies. Colonies were spotted on plates containing X-Gal. Interaction is visualized by the generation of a blue color. (E) Bacterial two-hybrid assay testing the interaction between RNE' and MinD proteins, either wild type and variants with point mutations. As in (F), with plasmids expressing RNE' together with WT or mutated MinD (L48R, V147R, L194R), used for the interaction studies. Interaction is visualized by the generation of a blue color. (F) MinD point mutations do not cause alteration in RNase E cellular localization. Images showing live cells deleted for *minD*, expressing RNase E-mCherry from the chromosome and MinD or its variants (MinD^L48R, MinD^V147R or MinD^L194R) from a plasmid. Scale bar corresponds to 5 μm. Source data are available online for this figure.

of RNA-loaded RNase E, given that MinD binding to RNase E is not expected to interfere with RNA binding. The results thus far suggest that, despite the lack of correlation in the subcellular trafficking of the two proteins, MinD affects RNase E localization, dynamics, and activity.

Finally, we asked if there is a bilateral effect on localization or oscillation of MinD by RNase E. To answer this question, we transformed *rne3071* cells with a plasmid expressing mGFP-MinD and followed MinD dynamics at the permissive and restrictive temperatures. The results in Fig. 6F show that MinD oscillation was not affected at either temperature, that is, MinD dynamics was not affected by the absence of RNase E. These results indicate that MinD affects RNase E dynamics, but the opposite is not true.

**Physiological consequences of defects in mRNA localization to the poles**

In higher organisms, mislocalization of mRNA leads to several developmental disorders (Cody et al, 2013; Wang et al, 2016; Basyuk et al, 2021). To assess the consequence of mRNA mislocalization in bacteria, we looked at the localization of the polar CheA protein, previously shown to localize to the cell pole (Maddock et al, 1993; Ames et al, 2002; Sourjik, 2004), in Δ*minCDE* cells. The results in Fig. 7A show that, as opposed to wild-type cells, in which CheA localizes mainly to the poles, in Δ*minCDE* cells, CheA predominantly forms lateral clusters. Interestingly, over-expression of MinD also caused a similar phenotype of CheA mislocalization (Fig. 7B). This result implies that failure to localize the *cheA* mRNA to the poles, in cells lacking the Min system or overexpressing MinD, precludes polar mislocalization of its protein product.

Next, we inhibited cell division using cephalexin to check if the formation of CheA lateral clusters was due to the inhibition of cell division in Δ*minCDE* cells. The results in Fig. 7C show that the CheA protein formed distinct polar foci in the cephalexin-treated wild-type cells, ruling out the possibility that the lateral clusters in the Δ*minCDE* cells are due to inhibition of cell division.

The localization of CheA to the cell poles depends on the chemoreceptors (Maddock and Shapiro, 1993). To rule out the possibility that the formation of lateral clusters by CheA in Δ*minCDE* cells is due to a defect in chemoreceptor localization, we monitored the localization of Tar-YFP in these cells. We did not find a notable difference between the localization of Tar-YFP in wild type and in Δ*minCDE* cells (Fig. 7D), nor in cells overexpressing MinD (Fig. 7E). Finally, we checked the localization of

the polar protein EI, whose transcript does not localize to the poles, in cells lacking the Min system (Govindarajan et al, 2013; Lopian et al, 2010). The EI protein remained at the cell poles in Δ*minCDE* cells (Fig. 7F). Altogether, our results suggest that the lateral clusters formed by the CheA protein in Δ*minCDE* strain are the result of mislocalization of the *cheA* mRNA in these cells.

# Discussion

The field of bacterial RNA localization is in its infancy and therefore, very little is known about the mechanisms of RNA localization in bacteria. Co-translational targeting of mRNAs to the membrane by the Signal Recognition Particle (SRP) has been documented in bacteria as a mechanism for proper localization of membrane proteins (Moffitt et al, 2016; Saraogi and Shan, 2014). Unlike the SRP-mediated localization of membrane transcripts, necessitating translation as an initial step to deliver the transcripts, we have shown that transcripts are capable of localizing to subcellular sites within the bacterial cells in a translation-independent manner (Nevo-Dinur et al, 2011). We later documented the subcellular localization of all mRNAs and small regulatory RNAs, using our Rloc-seq protocol, and showed that active translation is not obligatory for a significant fraction of the *E. coli* transcriptome in order to be targeted to different cellular domains, such as the membrane and the poles (Kannaiah et al, 2019). Very recently, mRNA targeting was shown to eliminate the need for SRP for membrane protein insertion in *E. coli* (Sarmah et al, 2023). These and additional studies published in recent years [reviewed in Kannaiah and Amster-Choder (2016) and Fei and Sharma (2018)] suggest that bacterial RNA localization is a regulated process and that mechanisms for localizing transcripts should exist in bacteria.

In this study, we identified cellular constituents that are required for the localization of polar transcripts. Using mutational study and microscopy, we discovered that the localization of polar transcripts is altered in Δ*minCDE* cells. Furthermore, MinD was detected as co-purifying with polar mRNA by mass spectrometry. MinD, together with MinC and MinE, are involved in determining proper placement of the division septum in *E. coli* by inhibiting its formation elsewhere (de Boer et al, 1989). This is the best-studied function of the Min proteins and, thus far, most studies have focused on this role and on their interaction with the divisome proteins. However, the Min system, like other systems and factors that work together to position the FtsZ-ring in the mid-cell, is not

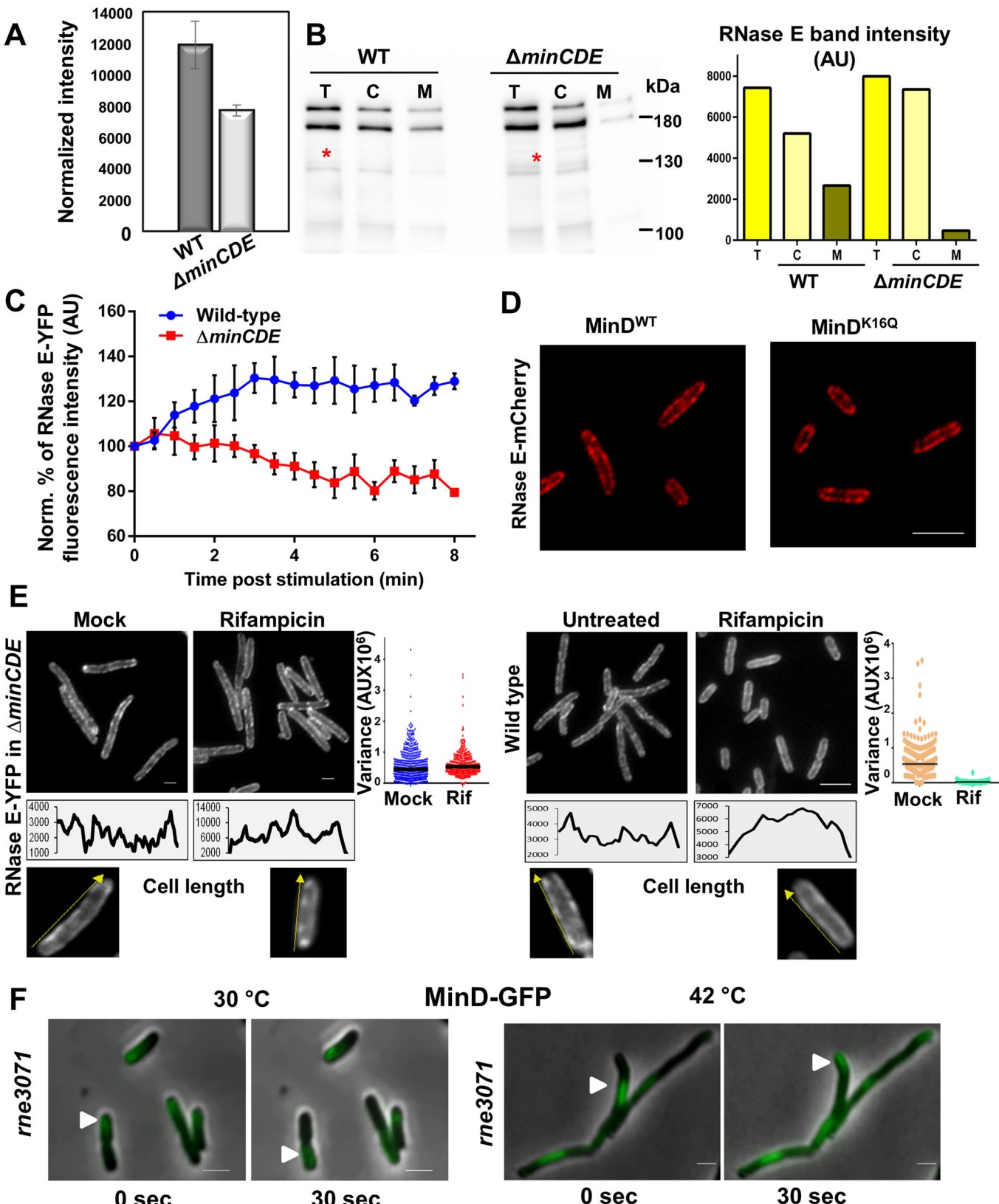

**Figure 6.  MinD is involved in membrane localization and dynamics of RNase E.**

(A) Relative quantification of RNase E levels in the membrane fraction after cell fractionation. RNase E fractionated 30% less efficiently in ∆*minCDE* cells than in wild-type cells. Data are the mean ± SD of biological triplicates. (B) Upper panel: Western blot analysis detecting RNase E in ∆*minCDE* compared to wild-type cells. An equal number of cells were fractionated to cytoplasm and membrane. The membrane was probed with anti-RNase E Ab (red asterisk). Lower panel: quantification of the bands in the Western blot analysis. T total, C cytoplasm, M membrane. (C) Mean YFP fluorescence signal was normalized (Norm.) to the background after photobleaching. Lines represent the recovery of fluorescent signals in wild-type (blue) and Min deleted (red) cells over time. Data are the mean ± SD of 15 biological replicates. (D) Images showing live cells deleted for *minD*, expressing RNase E-mCherry from the chromosome and wild-type MinD (right) or its MinD$^{K16Q}$ variant from a plasmid. (E) Images of RNase E-YFP in ∆*minCDE* (left panel) and wild type (right panel) cells untreated and treated with rifampicin. Scale bar corresponds to 2 µm. For each sample, an image that shows a magnified representative cell, as well as a graph showing the average fluorescence intensity profile of RNase E-YFP plotted against the long cell axis after normalizing to cell length, are shown below; *n* = 50–60. On the right of each strain, a variance extracted from the periphery of the cells from the image files is presented; untreated and rifampicin-treated cells showed comparable levels of variance. (F) MinD oscillation was not affected in RNase E temperature-sensitive mutant. Snapshots of MinD oscillation taken every 30 s in *rne3071* at the permissive (*upper panel*) or restrictive (*lower panel*) temperature. White arrowheads point at the oscillating MinD. Scale bar corresponds to 2 µm. Source data are available online for this figure.

essential for cell division. In line with that, several recent studies have speculated that the Min proteins play important roles in cellular processes other than cell division, such as RNA decay, bacterial motility, protein secretion, and positioning and regulation of inner membrane proteins [(reviewed in Taviti and Beuria (2019) and Ramm et al (2019)]. Relevant to our work, MinD was shown to interact directly with RNase E (Taghbalout and Rothfield, 2007), which, together with the other degradosome components, was identified by us as also required for the localization of our representative polar transcripts. Our findings imply that the capacity of MinD to oscillate from pole-to-pole to position the division machinery in mid-cell is exploited to place RNA at the two ends of the path, suggesting that during evolution the ability of the Min proteins to oscillate has been utilized to move "passengers" along the long cell axis, which include mRNA-protein complexes (mRNPs).

RNase E has emerged as the chief player in RNA processing and decay in *E. coli* and in many other bacteria (Mackie, 2013). RNase E was suggested to form helical structures, which are characteristic of cytoskeletal proteins (Taghbalout and Rothfield, 2007). Later studies suggested that although RNase E forms clusters that distribute along a helical path near the membrane, it does not form a cytoskeletal-like structure (Strahl et al, 2015a). Curiously, RNase E in *Caulobacter* cells was shown to form liquid-liquid phase-separated condensates, which compete with the ribosomes for mRNA substrates and affect decay during normal growth (Al-Husini et al, 2018). In nitrogen-starved *E. coli* cells, RNase E forms big clusters with Hfq at the poles (McQuail et al, 2022), although in other stresses, e.g., osmotic stress and late stationary phase, the *E. coli* RNase E does not localize with the Hfq polar clusters and remains distributed in small transiently-forming puncta on the inner membrane (Goldberger et al, 2022). It is worth mentioning that the distribution of the ribosomes and RNase E differs between the two bacterial species—near the membrane and at the poles in *E. coli*, and in the cytoplasm along the long cell axis in *Caulobacter* (Bayas et al, 2018; Gray et al, 2019; Carpousis et al, 2022). Specific localization of RNase E and its complex arrangement are supposed to have explicit implications on which population of transcripts is being degraded. If localized transcription occurs in *E. coli*, as suggested by the significant correlation between transcriptome and proteome localization (Kannaiah et al, 2019) and by the result in Fig. 7, then RNase E localization plays an indirect role in protein localization.

Although the interaction of RNase E with MinD has been reported (Taghbalout and Rothfield, 2007), its significance remained an open question. Our study provides an insight into

the roles of this interaction. Not only do we show that in the absence of MinD, RNase E is targeted less efficiently to the membrane and accumulates at the cell poles, but we also show that in such cells RNase E dynamics is halted and its ability to release transcripts is impaired. Hence, our results imply that MinD affects RNase E localization, dynamics and activity and is, thus, expected to influence which mRNAs will be degraded and where. As expected, the fraction of RNase E that localizes to the membrane is reduced, but not abolished in cells deleted for the Min proteins (see Fig. 6), as it conducts regulated degradation of transcripts other than the polar ones. We further show that not only do polar transcripts mislocalize in the absence of either MinD or RNase E, but these transcripts are also destabilized in cells that accumulate RNase E at the poles due to a lack of MinD, substantiating the role of MinD–RNase E interaction in determining and maintaining the polar transcriptome. The predicted utilization of the same domain in RNase E for binding to the membrane and to MinD should be further investigated in the future in order to elucidate the mechanism that connects MinD and RNase E to the membrane and its implications on RNA localization and potentially other cellular processes, such as translation. Noteworthily, the formation of RNase E puncta on the inner cytoplasmic membrane was recently shown to depend on an interaction with polyribosomes (Hamouche et al, 2021).

How can a major ribonuclease be in charge of transcripts positioning? The answer probably lies in the milieu in which it meets the RNAs. RNase E, like various eukaryotic ribonucleases, such as Xrn1 is present in mRNPs, whose paradigms are P-bodies and stress granules (Corbet and Parker, 2019). For many years, the presumed role of P-bodies was RNA degradation, but for quite a while now it is obvious that they play an important role in translation. The dynamic process, wherein mRNPs can move between polysomes, P-bodies and stress granules controls the balance between translation and mRNA degradation, which ensures the correct level of transcripts (Decker and Parker, 2012; Matheny et al, 2019). The competition between translation and mRNA degradation is best understood through changes in proteins that occupy the mRNPs assemblies, many of them already proved to form via phase separation, e.g., RNase E in *C. crescentus*, as well as P-bodies and stress granules in eukaryotic cells (Al-Husini et al, 2018; Corbet and Parker, 2019). Hence, auxiliary proteins that bind to transcripts and/or to RNase E are expected to determine the fate these transcripts when they encounter RNase E. Based on our results, which show that MinD affects the activity and dynamics of RNase E and its interaction with the membrane, and on previous findings, which showed that MinD affects MinE and MinC activity

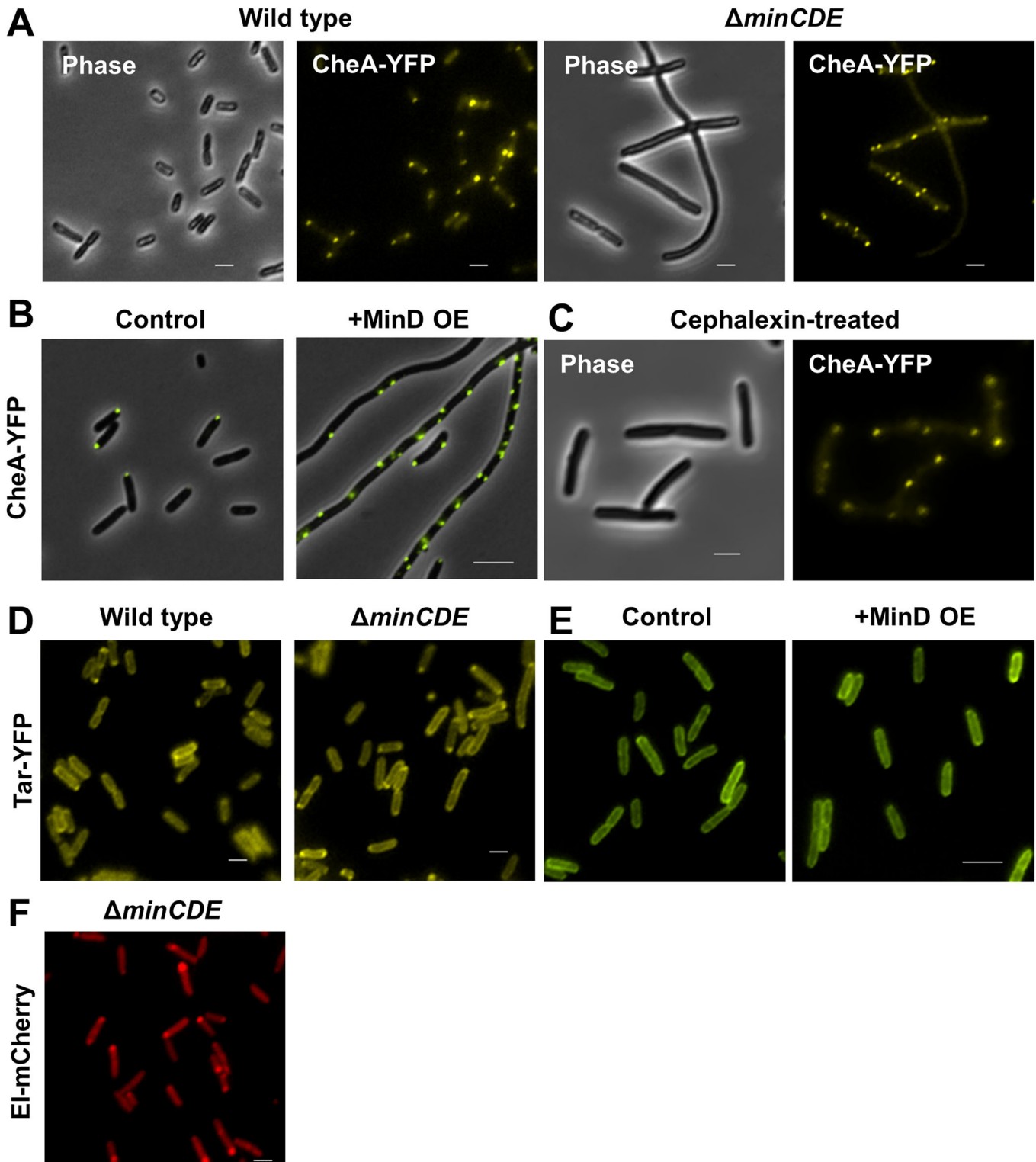

**Figure 7. Mislocalization of *cheA* mRNA leads to altered protein localization.**

(A) Images showing CheA-YFP protein in wild-type and Δ*minCDE* cells. (B) Images showing live cells expressing CheA-YFP from the chromosome with or without overexpression of MinD. (C) Images showing CheA-YFP protein in cephalexin-treated cells. (D) Images showing Tar-YFP protein in wild type and Δ*minCDE* cells. (E) Images showing live cells expressing Tar-mYFP from a plasmid with or without overexpression of MinD. (F) An images showing EI-mCherry protein in Δ*minCDE* cells. In (A–C), cells are also shown as phase-contrast images (gray). Scale bar corresponds to 2 μm. Source data are available online for this figure.

(Wu et al, 2011), MinD may very well affect RNase E localization and mode of action when it encounters polar transcripts that are "tagged" by auxiliary proteins. Whether these auxiliary proteins recognize MinD or affect the MinD-RNase E interplay remains to be investigated.

Finally, our results show that mislocalization of polar transcripts has a direct consequence on the proper localization of their encoded proteins. Taken together with our previous results, which showed a high correlation between the localization of mRNAs and their encoded proteins, as well as localization of a significant fraction of mRNAs independent of translation in *E. coli* (Kannaiah, 2019), mRNA localization seems to serve as a mechanism to localize proteins in bacteria. The extent of localized translation in bacteria and the detailed mechanism remain to be discovered.

# Methods and protocols

## Reagents and tools

See Table 1.

**Table 1.   Reagents and tools.**

| Reagent/resource | Reference or source | Identifier or catalog number |
|---|---|---|
| Antibodies | | |
| The Flag Tag Antibody, mAb, Mouse | GenScript | Cat#A00187 |
| Anti RNase E | Carpousis lab | N/A |
| Anti GroEL | Abcam | Cat#ab90522 |
| Chemicals | | |
| Ampicillin | AppliChem Panreac | Cat#A0389 |
| Chloramphenicol | Sigma-Aldrich | Cat #C0378 |
| Kanamycin sulfate | Biological Industries | Cat#25389-94-0 |
| Rifampicin | Goldbio | Cat#R-120-5 |
| GFP-TRAP | Chromotek | Cat#gta-20 |
| TriReagent Solution | Sigma-Aldrich | Cat#T9424 |
| DMSO | Sigma Aldrich | Cat#D2438 |
| X-Gal | Ornat | Cat#1758-0300 |
| InstantBlue® Protein Stain | Expedeon | Cat#ISB1L |
| IPTG | Ornat | Cat#1758-1400 |
| Recombinant Ribonuclease Inhibitor | Takara | Cat#2313A |
| Poly-L-lysine | Sigma | Cat#P8920 |
| Commercial assays | | |
| Direct-zol RNA Miniprep Kit | Zymo Research | Cat#R2060 |
| EZ-ECL | Biological Industries | Cat#20-500-120 |

**Table 1.**   (continued)

| Reagent/resource | Reference or source | Identifier or catalog number |
|---|---|---|
| Anti-DYKDDDDK G1 Affinity Resin | GenScript | Cat# L00432 |
| Turbo DNA-free Kit | Ambion | Cat#AM1907 |
| Gravity column | Bio-Rad | Cat#732-1010 |
| RevertAid RT Reverse Transcription Kit | Thermo Scientific | Cat#K1691 |
| ABsolute qPCR SYBR Green mix | Thermo Scientific | Cat#AB1158 |
| Software | | |
| NIS Elements Advanced Research (AR) version 4.5 | Nikon | N/A |
| GraphPad Prism v6 | GraphPad | https://www.graphpad.com/scientific-software/prism/ |

## Bacterial growth

Unless otherwise stated, cultures were grown in LB with appropriate antibiotics at 30 °C. TB was used for time-lapse microscopy. Kanamycin (30 µg/ml), ampicillin (100 µg/ml), chloramphenicol (25 µg/ml), or tetracycline (20 µg/ml) were added when appropriate. Unless otherwise stated, induction using IPTG was done at a final concentration of 0.1 mM.

## Strain and plasmid construction

All strains, plasmids, and oligonucleotides used in this study are listed in Tables EV1–EV3, respectively. To construct single gene deletions in wild-type strain, single gene deletion linked to kanamycin resistance gene was transferred from the Keio collection to wild-type cells by P1 transduction. The kanamycin resistance gene was removed by transforming the deletion strains with pCP20 and selecting on LB plates containing ampicillin at 30 °C. Plasmid pCP20 was later cured by growing the cells at 42 °C. MG1655 *ΔminCDE::kan* was constructed by P1 transduction of *ΔminCDE::kan* from PB114 to MG1655. To construct MG1655 *ΔmimD*, the kanamycin cassette from pKD4 plasmid was amplified with primers that contained the chromosomal sequences that flanked *minD*. The amplified fragment was introduced into MG1655 strain containing pKD46. The verified cells were grown at 42 °C and tested for loss of pKD46. pCP20 plasmid was used to remove the kanamycin cassette.

The construction of plasmids expressing transcripts tagged with MS2 binding sites (6xbs) was constructed as described previously (Kannaiah et al, 2019).

pET15b-FLAG-minD plasmid was constructed by amplifying the fragment that encodes the *minD* gene from the chromosome, using F-insert and R-insert primers at the start and the end of the *minD* gene. Both primers contained homology to the pET15b vector sequence. The vector was amplified using F-vector pet15b and R-vector pet15b primers followed by DpnI digestion. FLAG-minD insert was ligated to pET15b vector using Gibson assembly.

To construct the RNE'- and RNE'ΔA-overexpressing plasmids, the following sequences were synthesized in vitro (Hylabs):

RNE'-ATATACCATGGGCAGCAGCCATCATCATCATCATC
ACAGCAGCGGCCTGGTGCCGCGCGGCAGCCATATTTCTC
GCTTTGGCCTGCTGGAAATGTCCCGTCAGCGCCTGAGCCC
ATCACTGGGTGAATCCAGTCATCACGTTTGTCCGCGTTGTT
CTGGTACTGGCACCGTGCGTGACAACGAATCGCTGTCGCT
CTCTATTCTGCGTCTGATCGAAGAAGAAGCGCTGAAAGAG
AACACCCAGGAAGTTCACGCCATTGTTCCTGTGCCAATCG
CTTCTTACCTGCTGAATGAAAAACGTTCTGCGGTAAATGC
CATTGAAACTCGTCAGGACGGTGTGCGCTGTGTAATTGTG
CCAAACGATCAGATGGAAACCCCGCACTACCACGTGCTGC
GCGTGCGTAAAGGGGAAGAAACCCCAACCTTAAGCTACAT
GCTGCCGAAGCTGCATGAAGAAGCGATGGCGCTGCCGTCT
GAAGAAGAGTTCGCTGAACGTAAGCGTCCGGAACAACCTG
CGCTGGCAACCTTTGCCATGCCGGATGTGCCGCCTGCGCC
AACGCCAGCTGAACCTGCCGCGCCTGTTGTAGCTCCAGCA
CCGAAAGCTGCACCGGCAACACCAGCAGCTCCTGCACAAC
CTGGGCTGTTGAGCCGCTTCTTCGGCGCACTGAAAGCGCT
GTTCAGCGGTGGTGAAGAAACCAAACCGACCGAGCAACC
AGCACCGAAAGCAGAAGCGAAACCGGAACGTCAACAGGA
TCGTCGCAAGCCTCGTCAGAACAACCGCCGTGACCGTAAT
GAGCGCCGCGACACCCGTAGTGAACGTACTGAAGGCAGC
GATAATCGCGAAGAAAACCGTCGTAATCGTCGCCAGGCA
CAGCAGCAGACTGCCGAGACGCGTGAGAGCCGTCAGCAG
TAACATATGCCTCGAG,

RNE'ΔA-ATATACCATGGGCAGCAGCCATCATCATCATCA
TCACAGCAGCGGCCTGGTGCCGCGCGGCAGCCATATTTCT
CGCTTTGGCCTGCTGGAAATGTCCCGTCAGCGCCTGAGCC
CATCACTGGGTGAATCCAGTCATCACGTTTGTCCGCGTTGT
TCTGGTACTGGCACCGTGCGTGACAACGAATCGCTGTCGC
TCTCTATTCTGCGTCTGATCGAAGAAGAAGCGCTGAAAGA
GAACACCCAGGAAGTTCACGCCATTGTTCCTGTGCCAATC
GCTTCTTACCTGCTGAATGAAAAACGTTCTGCGGTAAATG
CCATTGAAACTCGTCAGGACGGTGTGCGCTGTGTAATTGT
GCCAAACGATCAGATGGAAACCCCGCACTACCACGTGCTG
CGCGTGCGTAAAGGGGAAGAAACCCCAACCTTAAGCTACA
TGCTGCCGAAGCTGCATGAAGAAGCGATGGCGCTGCCGTC
TGAAGAAGAGTTCGCTGAACGTAAGCGTCCGGAACAACCT
GCGCTGGCAACCTTTGCCATGCCGGATGTGCCGCCTGCGC
CAACGCCAGCTGAACCTGCCGCGCCTGTTGTAGCTCCAGC
ACCGAAAGCTGCACCGGCAACACCAGCAGCTGAAGAAAC
CAAACCGACCGAGCAACCAGCACCGAAAGCAGAAGCGA
AACCGGAACGTCAACAGGATCGTCGCAAGCCTCGTCAG
AACAACCGCCGTGACCGTAATGAGCGCCGCGACACCCGTA
GTGAACGTACTGAAGGCAGCGATAATCGCGAAGAAAACC
GTCGTAATCGTCGCCAGGCACAGCAGCAGACTGCCGAGA
CGCGTGAGAGCCGTCAGCAGTAACATATGCCTCGAG

The in vitro synthesized sequences were cleaved by NcoI and NdeI and ligated to the pET15b plasmid cleaved by the same enzymes to yield plasmids pET15b-His-RNE' and pET15b-His-RNE'ΔA, respectively.

To construct pUT18C-rne$^{378-659}$ and pUT18C-rne$^{378-659}$ΔA plasmids, the *in vitro* synthesized sequences mentioned above were amplified using primers F-PstI-rne-pUT18C and R-BamHI-rne to introduce PstI and BamHI sites at the 5' and 3' ends, respectively. The amplified fragments were digested using PstI and BamHI enzymes and ligated to pUT18C plasmid, which was previously cleaved by PstI and BamHI.

The pKT25-minD plasmid was constructed by amplifying the *minD* gene from MG1655 chromosome using primers F-PstI-minD

pKT25 and R-BamHI minD that introduced PstI and BamHI sites at the 5' and 3' ends, respectively. The amplified fragment was digested using PstI and BamHI enzymes and ligated to pKT25, which was previously cleaved by PstI and BamHI.

To construct MinD$^{L48R}$- and MinD$^{V147R}$-expressing plasmids, the following sequences were synthesized in vitro (Hylabs):

MinD$^{L48R}$-GCTGCACGCATTATTGTTGTTACTTCGGGCAA
AGGGGGTGTTGGTAAGACAACCTCCAGCGCGGCCATCGC
CACTGGTTTGGCCCAGAAGGGAAAGAAAACTGTCGTGAT
AGATTTTGATATCGGCCTGCGTAATCTCGAC**cgt**ATTATGGG
TTGTGAACGCCGGGTCGTTTACGATTTCGTCAACGTCATT
CAGGGCGATGCAACGCTAAATCAGGCGTTAATTAAAGAT
AAGCGTACTGAAAATCTCTATATTCTGCCGGCATCGCAAA
CACGCGATAAAGATGCCCTCACCCGTGAAGGGGTCGCCAA
AGTTCTTGATGATCTGAAAGCGATGGATTTTGAATTTATC
GTTTGTGACTCCCCGGCAGGGATTGAAACCGGTGCGTTA
ATGGCACTCTATTTTGCAGACGAAGCCATTATTACCACCA
ACCCGGAAGTCTCCTCAGTACGCGACTCTGACCGTATTTT
AGGCATTCTGGCGTCGAAATCACGCCGCGCAGAAAATGG
CGAAGAGCCTATTAAAGAGCACCTGCTGTTAACGCGCTA
TAACCCAGGCCGCGTAAGCAGAGGTGACATGCTGAGCATG
GAAGATGTGCTGGAGATCCTGCGCATCAAACTCGTCGGC
GTGATCCCAGAGGATCAATCAGTATTGCGCGCCTCTAAC
CAGGGTGAACCGGTCATTCTCGACATTAACGCCGATGCGG
GTAAAGCCTACGCAGATACCGTAGAACGTCTGTTGGGAG
AAGAACGTCCTTTCCGCTTCATTGAAGAAGAGAAGAAAGG
CTTCCTCAAACGCTTGTTCGGAGGAtaa

MinD$^{V147R}$- GCTGCACGCATTATTGTTGTTACTTCGGGCAA
AGGGGGTGTTGGTAAGACAACCTCCAGCGCGGCCATCGCC
ACTGGTTTGGCCCAGAAGGGAAAGAAAACTGTCGTGATAG
ATTTTGATATCGGCCTGCGTAATCTCGACCTGATTATGGGT
TGTGAACGCCGGGTCGTTTACGATTTCGTCAACGTCATTC
AGGGCGATGCAACGCTAAATCAGGCGTTAATTAAAGATA
AGCGTACTGAAAATCTCTATATTCTGCCGGCATCGCAAAC
ACGCGATAAAGATGCCCTCACCCGTGAAGGGGTCGCCAA
AGTTCTTGATGATCTGAAAGCGATGGATTTTGAATTTATC
GTTTGTGACTCCCCGGCAGGGATTGAAACCGGTGCGTT
AATGGCACTCTATTTTGCAGACGAAGCCATTATTACCACC
AACCCGGAA**cgt**TCCTCAGTACGCGACTCTGACCGTATTTT
AGGCATTCTGGCGTCGAAATCACGCCGCGCAGAAAATGGC
GAAGAGCCTATTAAAGAGCACCTGCTGTTAACGCGCTATA
ACCCAGGCCGCGTAAGCAGAGGTGACATGCTGAGCATGG
AAGATGTGCTGGAGATCCTGCGCATCAAACTCGTCGGCGT
GATCCCAGAGGATCAATCAGTATTGCGCGCCTCTAACCAG
GGTGAACCGGTCATTCTCGACATTAACGCCGATGCGGGT
AAAGCCTACGCAGATACCGTAGAACGTCTGTTGGGAGAA
GAACGTCCTTTCCGCTTCATTGAAGAAGAGAAGAAAGGC
TTCCTCAAACGCTTGTTCGGAGGAtaa

The in vitro synthesized sequences were amplified using F-26bp(KpnI)-minD and R-minD-25bp(EcoRI) primers to introduce KpnI and EcoRI restriction sites, respectively. The pKT25 plasmid was amplified using F-pkT25(EcoRI) and R-pkT25(KpnI) primers, followed by DpnI digestion. Both inserts were digested using KpnI and EcoRI and ligated to the amplified pKT25 plasmid cleaved by the same enzymes to yield plasmid pKT25-minD$^{L48R}$ and pKT25-minD$^{V147R}$.

A plasmid expressing MinD$^{L194R}$ fused to the T25 domain of adenylate cyclase protein (Karimova et al, 2017) was constructed using inverse PCR reaction on The pKT25-minD plasmid. The

F-L194Rnew (phosphorylated) and R-L194R primers were used to amplify the plasmid, followed by self-ligation to create pKT25-minD$^{L194R}$.

## Fluorescence microscopy

MG1655 was transformed with the required plasmids. Cells were grown overnight in LB with appropriate antibiotics at 30 °C, diluted 100-fold in fresh LB with appropriate antibiotics, and grown to exponential phase. The MS2-GFP protein was induced by adding 0.1–0.4% arabinose for 1–2 h and MS2 binding sites-tagged transcripts were induced by adding 0.1–0.2 mM IPTG for 1–1.5 h. To arrest transcription, 2 mg/ml of rifampicin was added for the last 10 min of growth. To obtain spherical or filamentous *E. coli* cells, A22 (2 µg/ml) or cephalexin (20 µM) were added from the beginning of the growth, respectively. To visualize mRNA localization in the temperature-sensitive mutant of RNase E (*rne3071*), overnight cells grown at 30 °C were diluted 100-fold in fresh LB with appropriate antibiotics and grown to OD$_{600}$ = 0.15 at 30 °C; cultures were either shifted to the restrictive temperature (42 °C) or continued to grow at 30 °C for 2 h. To visualize mRNA localization in the enolase mutant (*eno-2*), cells were grown in M9 succinate medium (0.4%) at 30 °C. Expression of the MS2-GFP and MS2 binding sites-tagged transcripts was induced for 1.5 h prior to collection of cells. For viewing mRNAs and proteins, cells were placed on a slide with poly L lysine-coated coverslips or agarose pad with uncoated coverslips, respectively. Cells were imaged using Nikon Eclipse Ti-E inverted microscope equipped with Perfect Focus System (PFS) and ORCA Flash 4 camera (Hamamatsu photonics). Images were processed, and two-dimensional (2D) deconvolution was performed using NIS Elements-AR software.

## Time-lapse microscopy

Time-lapse imaging was performed using Nikon Eclipse Ti-E equipped with OKOLAB cage incubator. To monitor RNase E localization, an overnight culture grown in TB was diluted 1:100 in fresh TB and cells were allowed to grow at 30 °C until they reached OD$_{600}$ of 0.2. Samples were then spotted on 1% TB agarose pads, which had been preequilibrated to 30 °C, and imaged immediately by time-lapse microscopy. To monitor MinD oscillation, cells expressing mGFP-MinD were grown overnight in LB and diluted in fresh LB and allowed to grow until they reached OD$_{600}$ of 0.4. IPTG (0.1 mM) was added for the last 20 min of growth. To monitor MinD oscillation under the restrictive condition in *rne-3017* mutant, an overnight culture grown in LB was diluted in fresh LB and allowed to grow until it reached the exponential phase at 30 °C. The culture was then shifted to 42 °C and allowed to grow for 1 h. The cells were spotted on 1% TB agarose pads, and images were acquired every 30 s for a total of 5 min.

## Fluorescence recovery after photobleaching (FRAP)

FRAP was performed as described previously (Kannaiah et al, 2019). Briefly, Nikon A1R confocal microscope equipped with Apochromat ×60 objective (numeric aperture 1.4) was used. Photobleaching was done over an area in which RNase E-YFP is localized. Recovery was measured every 30 s for a total period of

8 min. Mean fluorescence intensity for each region of intensity (ROI) was normalized to the background fluorescence intensity after bleaching. Images were analyzed using the NIS Elements AR module.

## Immunoprecipitation using GFP-Trap beads

### Cell lysis
Minicells were purified from *E. coli* χ1488 essentially as described previously (Lai et al, 2004). Briefly, cells grown to the exponential phase were harvested by low-speed centrifugation. The pellet, enriched with rod cells, was separated from the supernatant, and enriched with the minicells. The pellet was washed twice with 50 mM Tris-Cl to purify the rod cells. The supernatant was centrifuged at low speed for several times until there was no visible pellet. Finally, the minicells in the supernatant were pelleted. Purification of rod- and mini-cells was verified using microscopy. The pellet was resuspended in 1 ml of dilution buffer (10 mM Tris/Cl pH 7.5, 150 mM NaCl, 0.5 mM EDTA, 1 mM AEBSF added freshly) containing RNase inhibitor (0.1 unit/µl). The cells were broken by passing the suspension thrice through the microfluidizer at 12,000 lb/in².

### Binding
GFP-TRAP beads were equilibrated in a dilution buffer. In all, 20–30 µl bead slurry was resuspended in 500 µL dilution buffer (ice cold) and centrifuged at 4000 rpm for 2 min at 4 °C. The supernatant was discarded, and the beads were washed twice with 500 µl dilution buffer (ice cold). The cell lysate was added to the equilibrated GFP-TRAP beads and, after the addition of RNase inhibitor (0.1 unit/µl), incubated with gentle end-over-end mixing for 2 h at 4 °C. The tube was centrifuged at 4000 rpm for 2 min at 4 °C, and the supernatant was discarded. The pellet was washed twice with 500 µl dilution buffer (ice cold). The GFP-TRAP beads were resuspended in 50 µl dilution buffer.

## Reverse transcription PCR (RT-PCR)

RNA was isolated from the GFP-TRAP immunoprecipitated eluate. The concentration and purity of the RNA from the different samples were determined. The RNA samples were treated with DNase to remove DNA. DNA-free RNA was used for cDNA synthesis. The presence or absence of *bglG* transcripts in the immunoprecipitated eluate was analyzed using *bglG* primers.

## Protein analysis by LC-MS/MS and data processing

MS2-bound proteins were subjected to on-bead digestion with trypsin. The complexes were reduced with 1 mM Dithiothreitol for 30 min, followed by alkylation with 5 mM Iodoacetamide for 30 min in the dark. Proteins were digested overnight with trypsin (Promega) and digestion was terminated by adding 0.1% trifluoroacetic acid (TFA). The peptides were desalted and concentrated on C18 stage tips (Rappsilber et al, 2003). Prior to MS analysis, peptides were eluted from the stage tips using 80% acetonitrile, vacuum-concentrated and diluted in loading buffer (2% acetonitrile and 0.1% trifluoroacetic acid) and subjected to MS measurements. LC-MS/MS analysis was performed using nano-ultra high-performance liquid chromatography (nano-UPLC; Easy-nLC1000;

Thermo Scientific) coupled online to a Q-Exactive Plus mass spectrometer (Thermo Scientific). Peptides were eluted using 240-min gradient with acetonitrile and water. Raw files were analyzed with MaxQuant in-built with Andromeda search engine (Cox and Mann, 2008; Cox et al, 2011). MS/MS spectra were searched with reference to human UNIPROT database. False discovery rates were set for 0.01 for protein and peptide identification. All the statistical analyses of the MaxQuant output tables were performed with the Perseus software (Tyanova et al, 2016). Significant interactors of MS2-binding RNA sequence were examined by performing $T$ test for test transcripts against control transcripts with a $p$ value 0.01.

## Western blot analysis

Equal amounts of proteins were separated on semi-native polyacrylamide gels. Gels were subjected to Western blot analysis as described previously (Nevo-Dinur et al, 2011). The membrane was probed with α-RNaseE and α-GroEL (Abcam) for the detection of RNase E and GroEL, respectively. To detect MinD and its variants, α-MinD antiserum was used. Signals were visualized by the ECL system (Amersham).

## Cell fractionation

To fractionate cells to cytoplasm and membrane, overnight cultures were back-diluted 1:100 and grown to mid-exponential phase. An equal amount of cells from the different strains were then pelleted and immediately resuspended in native lysis buffer (50 mM $NaH_2PO_4$, 300 mM NaCl, 10 mM imidazole, pH = 8). Lysozyme and protease inhibitor were added followed by 30 min incubation on ice. The cells were then lysed using sonication, and a sample was taken to analyze the total cell lysate content. The lysate was centrifuged at 13,000 rpm for 10 min at 4 °C to separate the soluble cytoplasmic fraction (supernatant) from the membrane fraction (pellet).

## Quantification and statistical analysis

Western blot quantification was performed using ImageJ software. To generate average fluorescence intensity profiles, the fluorescence intensity along the long cell axis (see Fig. 1) was derived from NIS Elements-AR software, and the data were processed using MATLAB. All statistical analyses were performed using GraphPad Prism software (GraphPad).

## Identification of MinD-binding site on RNaseE using FlexPepBind

To identify the region in RNaseE that interacts with MinD, we first inspected the previously published structures of MinD to find a suitable template. The 2.6 A resolution crystal structure of MinD complexed with MinE (residues 12–31) peptide (PDB id: 3r9i) was selected and relaxed using the Rosetta FastRelax protocol (Nivon et al, 2013). The RNase sequence segment 378–737 was divided into overlapping sliding windows of 14 residues. Each sliding window was threaded on the MinE backbone of the relaxed template using Rosetta fixbb (Kuhlman et al, 2003) followed by full-atom flexible refinement using Rosetta FlexPepDock. The minimized models were analyzed using the pep_score_noref score that emphasizes the contribution of the peptide to binding.

## mRNA half-life measurement

MG1655 and MG1655 Δ*minCDE* cells were grown to an $OD_{600}$ of 0.18. Rifampicin (400 µg/ml) was then added for 15 min. Samples were collected every 3 min after the addition of rifampicin. Total RNA was isolated, and RNA concentrations were determined using a NanoDrop machine (NanoDrop Technologies). DNA was removed by DNase treatment and 1 µg of DNA-free RNA was used for cDNA synthesis. cDNA was quantified by real-time PCR using SYBR-green mix in Rotor Gene 3000A (Corbett) according to the manufacturer's instructions. Primers were designed for polar and non-polar mRNAs, and their expression was normalized using 16S rRNA levels. The relative amount of cDNA was calculated using the standard curve method obtained from PCR on serially diluted genomic DNA templates. The corrected values were plotted as a percent of the initial value versus time, and smoothed curves were fitted using an exponential function $I(t) = I0 \exp(-kt)$, where $I0$ is the initial mRNA concentration, $I(t)$ is the concentration after $t$ min, and $k$ is the decay coefficient in minutes−1, extracted from the exponential fit to the decay plot. The half-life was calculated from this decay coefficient using the relation $t1/2 = \ln(2)/k$.

## 9S precursor expression level measurement

The level of the 9S rRNA precursor of the p5S transcript was determined by qPCR as described above for the polar and non-polar mRNAs with two exceptions. First, cells were grown to the mid-exponential phase and then transferred to either permissive (30 °C) or restrictive (42 °C or 43 °C) temperature for 1 h of additional growth. Second, expression was normalized using *gyrA* mRNA as a reference.

## MinD protein purification

For the purification of FLAG-MinD, the protein was overexpressed in BL21(DE3) cells from the pET15b-FLAG-minD plasmid. Five hundred (500) ml cultures were harvested and lysed using glass beads and a Mixer Mill MM400 instrument. The lysates were loaded on an Anti-DYKDDDDK G1 Affinity Resin in a gravity column (Bio-Rad) and treated as suggested by the manufacturer. Proteins were eluted using incubation with 150 µg/ml Flag peptide for 30 min followed by centrifugation. The concentration and purification of MinD were verified using Bradford assay and Western blot analysis.

## Far-Western analysis

Far-Western analyses were carried out essentially as described previously (Nussbaum-Shochat and Amster-Choder, 1999). Over-night cultures of BL21(DE3). Cells overexpressing RNE' or RNE'ΔA fused to His tag from pET15b-rne[378-659]-His or pET15b-rne[378-659]ΔA-His, respectively, were diluted 1:100 in fresh LB and grown to mid-exponential phase, followed by induction with 1 mM IPTG. Cells were allowed to grow for 3 more hours. Subsequently, cells were pelleted, washed in 1× PBS, resuspended in Laemmli sample buffer, heated to 95 °C for 10 min and fractionated on a 15–20% SDS-polyacrylamide gel (Bio-Rad). The proteins were blotted onto a nitrocellulose membrane. The membrane was incubated with purified FLAG-MinD in 7.5% Difco skim milk (BD biosciences) diluted in 5 ml PBST (phosphate buffered saline

with Tween-20) for overnight at 4 °C, washed, incubated with anti-FLAG antibody for 2 h, washed trice and incubated with anti-rabbit antibody for 1 h. Proteins were verified for their level by Coomassie blue staining.

## Bacterial two-hybrid assay

Bacterial two-hybrid assays were carried out essentially as described previously (Karimova et al, 2000, 2017). Briefly, BTH101 strain was co-transformed with a combination of pKT25 and pUT18C expressing the proteins Zip, RNE', RNE'ΔA, MinD, MinD$^{L48R}$, MinD$^{V147R}$ or MinD$^{L194R}$, as indicated in the corresponding figure. Cells were grown for overnight in biological replicates in LB containing kanamycin, ampicillin and 0.5 mM IPTG. Interaction between the two hybrid proteins was monitored on LB X-GAL plates [LB agar supplemented with ampicillin (100 μg/ml), kanamycin (50 μg/ml), X-Gal (40 μg/ml), and IPTG (0.5 mM)].

## Data availability

The mass spectrometry proteomics data have been deposited to the ProteomeXchange Consortium via the PRIDE partner repository with the dataset identifier PXD044814.

## Peer review information

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

## Acknowledgements

We acknowledge Kenn Gerdes and Agamemnon Carpousis for the gift of strains and plasmids. We acknowledge Piet de Boer for the Rabbit anti-MinD antiserum and the MinD(K16Q) mutant. We acknowledge Ady Vaknin for the CheA-YFP expressing strain. We appreciate fruitful discussions with members of Orna Amster-Choder lab. Research in OA-C lab was supported by the Israel Science Foundation (ISF) founded by the Israel Academy of Sciences and Humanities (grant 1274/19). OA-C is an incumbent of the Dr. Jacob Grunbaum Chair in Medical Sciences. Research in the OS-F lab was supported by ISF (grant 301/2021).

## Author contributions

**Shanmugapriya Kannaiah**: Conceptualization; Data curation; Formal analysis; Validation; Investigation; Methodology; Writing—original draft; Writing—review and editing. **Omer Goldberger**: Conceptualization; Data curation; Formal analysis; Validation; Investigation; Visualization; Methodology; Writing—original draft; Writing—review and editing. **Nawsad Alam**: Data curation; Formal analysis; Validation; Investigation; Visualization; Methodology; Writing—original draft. **Georgina Barnabas**: Data curation; Software; Formal analysis; Validation; Investigation; Visualization; Methodology. **Yair Pozniak**: Data curation; Formal analysis; Validation; Investigation; Visualization; Methodology. **Anat Nussbaum-Shochat**: Data curation; Formal analysis; Validation; Investigation; Visualization; Methodology. **Ora Schueler-Furman**: Data curation; Software; Formal analysis; Funding acquisition; Validation; Visualization; Methodology. **Tamar Geiger**: Data curation; Software; Formal analysis; Funding acquisition; Validation; Visualization; Methodology. **Orna Amster-Choder**: Conceptualization; Formal analysis; Supervision; Funding acquisition; Validation; Investigation; Visualization; Methodology; Writing—original draft; Project administration; Writing—review and editing.

## Disclosure and competing interests statement

The authors declare no competing interests.

# Expanded View Figures

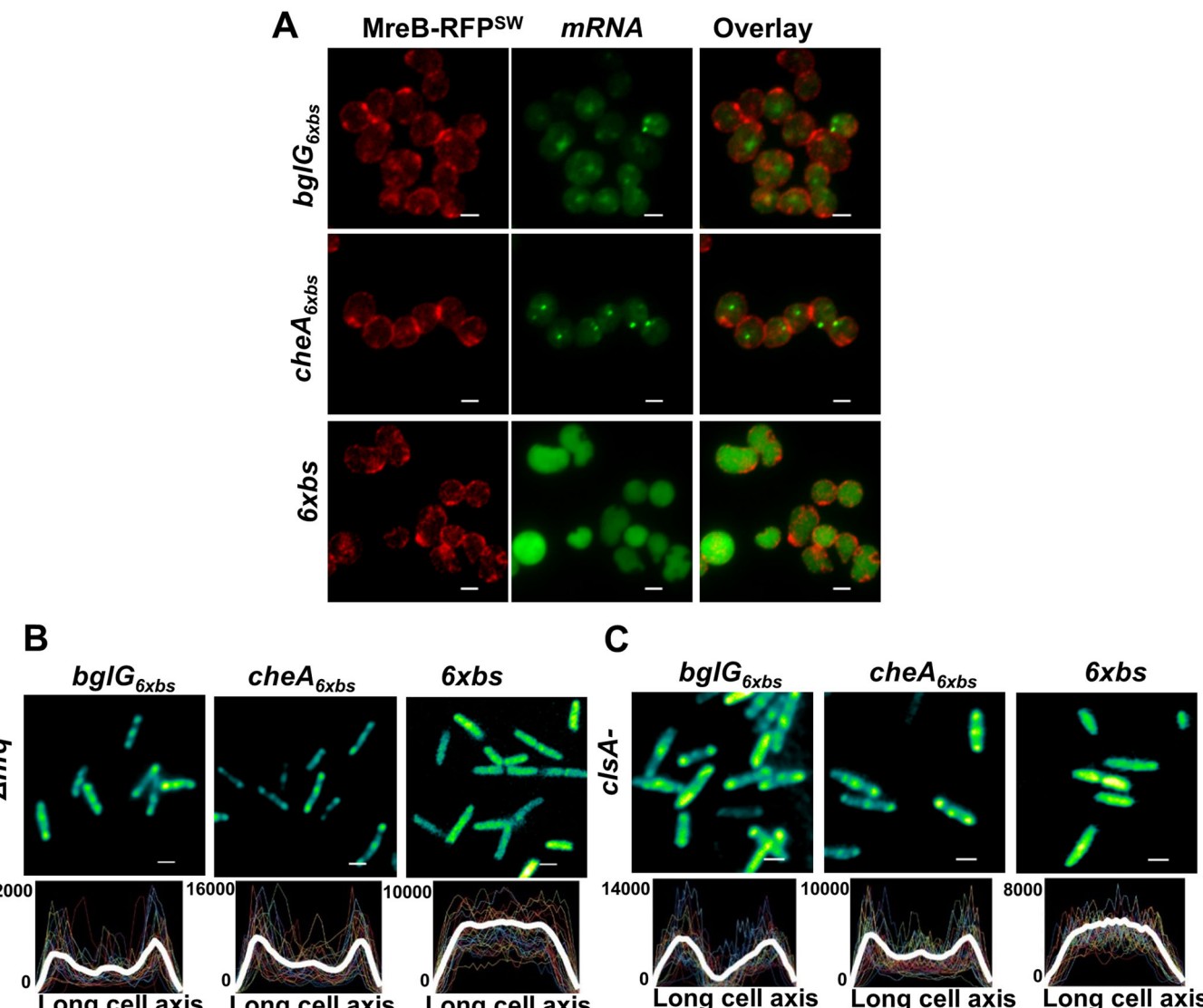

**Figure EV1. MreB, RNA chaperone Hfq, and cardiolipin are not involved in the localization of polar mRNAs (related to Fig. 1).**

(A) Images of *bglG*, *cheA* and 6xbs in A22-treated cells, which also express MreB-RFP$^{SW}$. Overlays of MreB (RFP) and mRNA (GFP) are also displayed. (B, C) Upper panels: Images showing localization of representative polar transcripts, *bglG*, *cheA*, as well as no mRNA control (6xbs) in Δ*hfq* and *clsA-* strain background, respectively. Lower panels: The average fluorescence intensity profiles of the transcripts plotted against the long cell axis after normalizing to cell length; *n* = 50–60 in both cases. mRNAs were detected in live cells by the MS2 system. Images are representatives of biological triplicates (A, B). Scale bar corresponds to 2 μm.

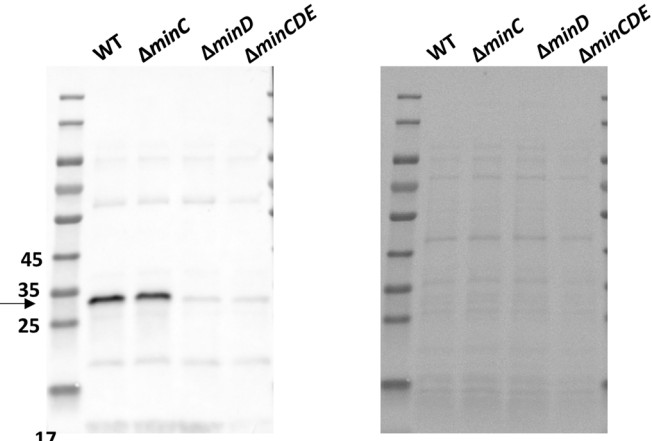

**Figure EV2.  Deletion of the *minC* gene does not affect the expression of *minD* (related to Fig. 1).**

Western blot analysis detecting MinD expression in log phase of Δ*minC*, Δ*minD* and Δ*minCDE* compared to wild type cells (left panel). An equal number of cells were sampled and blotted onto the membrane, as can be seen in the Ponceau S staining of the membrane before probing (right panel). The membrane was probed with anti-MinD antiserum. The band corresponding to MinD is indicated with an arrowhead (29.6 kDa).

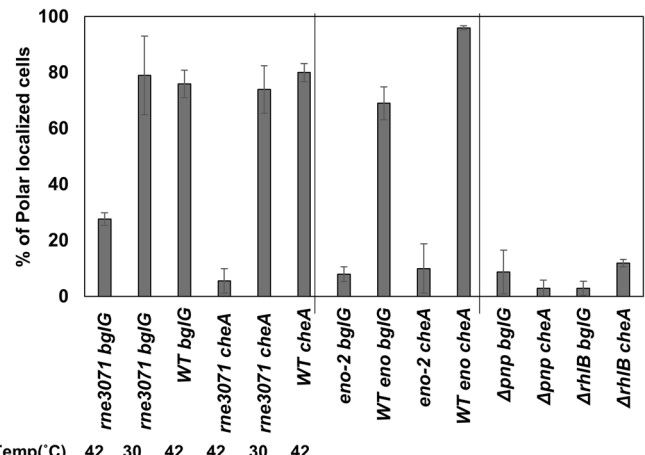

**Figure EV3.  Quantification of the spatial distribution of polar transcripts in cells deleted or disrupted for functional activity of degradosome components (related to Fig. 2).**

Bar plots showing the percentage of pole-localized transcripts in the different strains deleted or disrupted for the degradosome components in the experiment presented in Fig. 2. SEM error bars are presented for each sample. $n > 200$ for each strain.

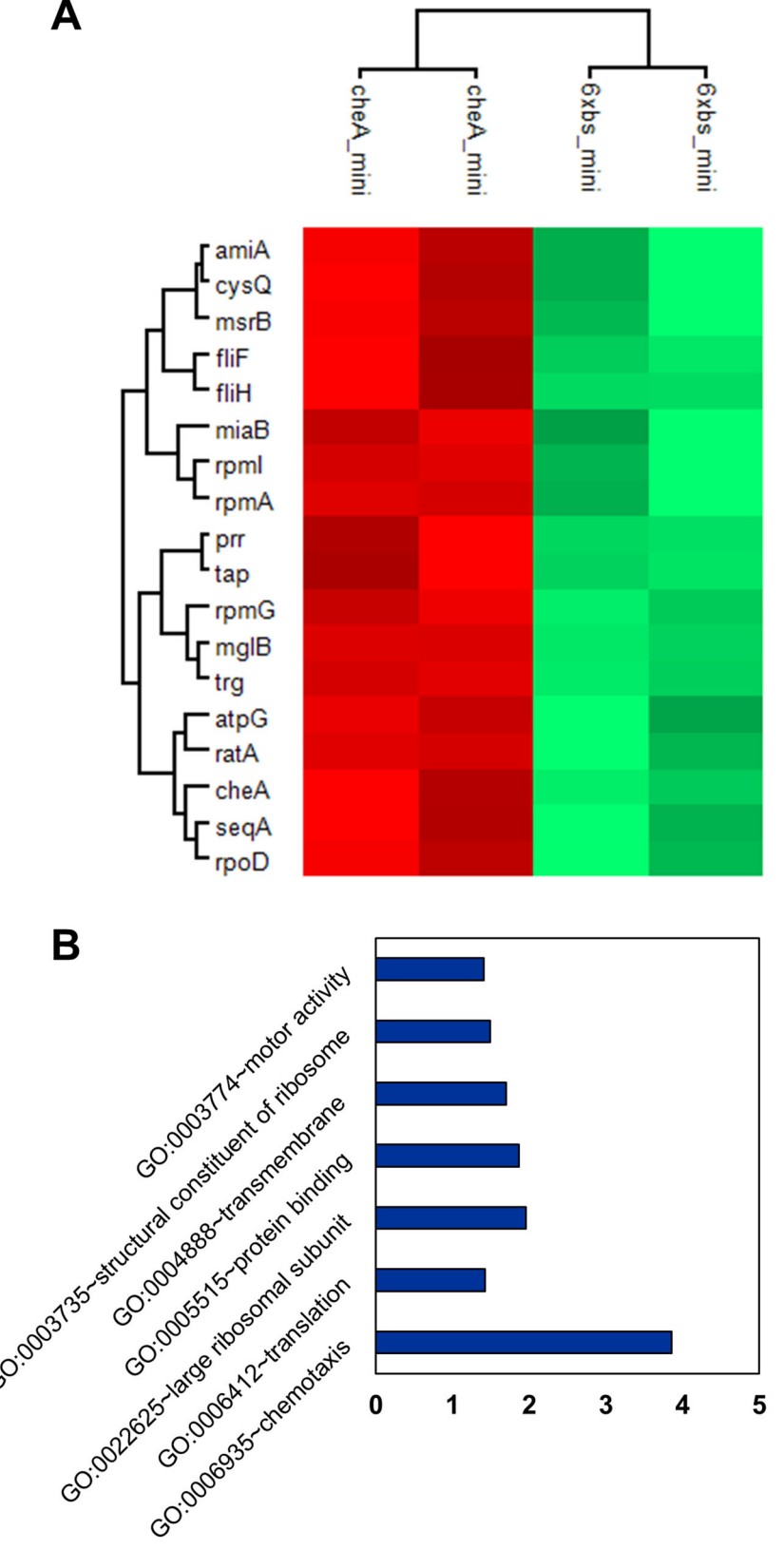

**A.** Heat map showing differentially enriched proteins that were pulled down with the *cheA* mRNA and with the no mRNA control (6xbs); p value<0.01.

**B.** Graphs showing statistically significant GO terms for the immunoprecipitated proteins that were identified by mass spectrometry as binding to the *cheA* RNA in minicells.

**Figure EV4. High-throughput proteomics screen to identify *cheA* mRNA-binding proteins (related to Fig. 3).**

(A) Heat map showing differentially enriched proteins that were pulled down with the *cheA* mRNA and with the no mRNA control (6xbs); *p* value < 0.01. (B) Graphs showing statistically significant GO terms for the immunoprecipitated proteins that were identified by mass spectrometry as binding to the *cheA* RNA in minicells.

**A**

**Induction of**
*minD* **expression:**          **-**                    **+**

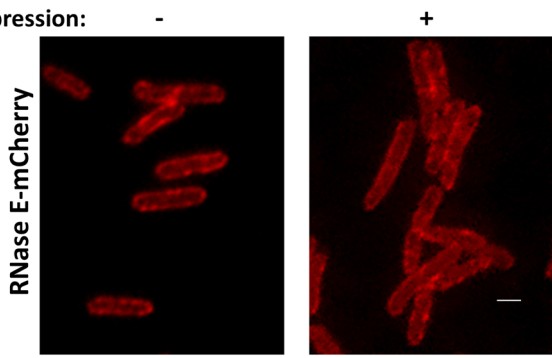

**B**    **Induction of**
       *minD* **expression:**      **-**      **+**

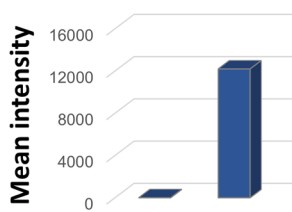

**Figure EV5.   MinD overexpression does not alter RNase E localization (related to Fig. 4).**

(**A**) Images showing live cells expressing RNase E-mCherry from the chromosome and MinD from a plasmid with or without induction. Scale bar corresponds to 2 μm. (**B**) Bar plot showing the mean intensity of MinD-GFP expression with or without induction.

