## [Peer Review File · The EMBO Journal]

MinD-RNase E interplay controls localization of polar mRNAs in *E. coli*

Shanmugapriya Kannaiah, Omer Goldberger, Nawsad Alam, Georgina Barnabas, Yair Pozniak, Anat Nussbaum-Shochat, Ora Schueler-Furman, Tamar Geiger, and Orna Amster-Choder

DOI: [10.15252/emj.2023113657](https://doi.org/10.15252/emj.2023113657)

Corresponding authors: Orna Amster-Choder (ornaam@ekmd.huji.ac.il) , Shanmugapriya Kannaiah (kannaiah@wustl.edu)

Review Timeline:

Submission Date:	3rd Feb 23
Editorial Decision:	21st Mar 23
Revision Received:	4th Sep 23
Editorial Decision:	4th Oct 23
Revision Received:	11th Dec 23
Accepted:	18th Dec 23

Editor: Ieva Gailite

Transaction Report:

Dear Dr. Amster-Choder,

Thank you for submitting your manuscript for consideration by the EMBO Journal. We have now received comments from three reviewers, which are included below for your information.

As you will see from the reports, the reviewers find the proposed role of MinD-dependent regulation of polar RNA stability via modulation of RNase E membrane localisation of interest. However, they also find that substantial additional evidence would be needed to convincingly support the proposed mechanism. If you find that you are able to address the issues raised by the reviewers, I would be happy to consider a revised version of the manuscript. I think it would be helpful to discuss the revision in more detail via email or phone/videoconferencing - please let me know which option you prefer. I should also add that it is The EMBO Journal policy to allow only a single major round of revision and that it is therefore important to resolve the main concerns at this stage.

We generally allow three months as standard revision time. As a matter of policy, competing manuscripts published during this period will not negatively impact on our assessment of the conceptual advance presented by your study. However, please contact me as soon as possible upon publication of any related work to discuss the appropriate course of action. Should you foresee a problem in meeting this three-month deadline, please let us know in advance and we may be able to grant an extension.

When preparing your letter of response to the referees' comments, please bear in mind that this will form part of the Review Process File and will therefore be available online to the community. For more details on our Transparent Editorial Process, please visit our website: <https://www.embopress.org/page/journal/14602075/authorguide#transparentprocess>. Please also see the attached instructions for further guidelines on preparation of the revised manuscript.

Please feel free to contact me if you have any further questions regarding the revision. Thank you for the opportunity to consider your work for publication. I look forward to discussing your revision.

Yours sincerely,

Ieva Gailite

- a point-by-point response to the referees' comments, with a detailed description of the changes made (as a word file).
- a word file of the manuscript text.
- individual production quality figure files (one file per figure)
- a complete author checklist, which you can download from our author guidelines

(<https://www.embopress.org/page/journal/14602075/authorguide>).
- Expanded View files (replacing Supplementary Information)
Please see out instructions to authors
<https://www.embopress.org/page/journal/14602075/authorguide#expandedview>

We realize that it is difficult to revise to a specific deadline. In the interest of protecting the conceptual advance provided by the work, we recommend a revision within 3 months (19th Jun 2023). Please discuss the revision progress ahead of this time with the editor if you require more time to complete the revisions.

Referee #1:

In this manuscript the authors present experimental work on the search for factors involved in the localization of mRNA to the poles of the rod-shaped *E. coli* cell. Two approaches were used. In the first, the polar localization of *bglB* and *cheA* mRNA was used to screen candidate factors such as cell division proteins. In the second, a proteomics approach was used to identify proteins that co-purified with *bglG*-MS2-GFP mRNA. The cell division protein MinD and the RNA degradosome component RNase E were identified as factors that affected the localization of *bglG* and *cheA* mRNA. Using a structure-based in silico approach, the authors predicted an interaction between the membrane attachment site of RNase E and MinD. The interaction was validated experimentally by Far Western blotting and two hybrid analyses including mutation of residues in MinD that were predicted to be necessary for the interaction with RNase E. Further experimental work investigated the effect of the deletion of the *minCDE* operon on the localization of RNase E, the *cheA* mRNA, and the TAR and EI proteins. Overall, the manuscript is logically organized and well written.

In the comments and questions, issues that need to be addressed by additional experimental work are marked with an asterisk. Comments.

C1) A strain and plasmid table indicating strain background, genotypes, and key plasmid features (backbone, ori, drug resistance and constructs for expression of mRNA and protein) needs to be added as part of the requirement that the work can be reproduced by other researchers in the field.

C2) Regarding the results in Fig. 3, assuming that the *bglG*-MS2-GFP mRNA is translated, I would have expected to see more ribosomal proteins. Also, what are possible explanations for the pulldown of two subunits of DNA dependent RNA polymerase? Please comment.

*Most importantly, this protocol should be validated by pulling down a second mRNA. For example, *cheA*-MS2-GFP.

C3) Fig. 4B and text. The conclusion that *minCDE* is responsible for mobility of RNase E is an overinterpretation of the FRAP results. If RNase E is detached from the membrane and trapped in the focus, then reduced diffusion is an indirect effect of the *minCDE* knockout. It is an error to suggest that *minCDE* actively promotes RNase E diffusion in the wild type cell.

C4) The data in Fig. EV7 and text. The conclusion that *cheA* mRNA is selectively destabilized in the *minCDE* mutant is an over interpretation of the experimental result.

*Additional stability measurements are required including at least one more mRNA localized to the cell pole (*bglB*?) as well as control mRNAs that are not localized to the cell pole.

C5) Line 900. The title is mislabeled. This is the legend for Fig. 5

C6) Labelling of Far Western blot in Fig. 4E is confusing. The heading 'Blot reacted with:' is not necessary since it is the same for both lanes. Also 'In gel lysate overexpressing:' is unnecessary. The panel could be simply labelled 'Far Western Blot'.

C7) There is a discrepancy in the labelling of Panels C and G in Fig. 4. Panel C indicates 3 leucine residues: L48, L147 and L198. Panel G indicates a V147R mutation. The text also mentions V147. Considering the model in Panel C, it looks like V147 is mislabeled. The arrow points to a leucine residue in segment-A of RNase E. I suspect that V147 is a MinD residue slightly to the left of the leucine in segment-A of RNase E.

C8) Line 304. The citation of McQuail et al. 2022 here is misleading. The large focus of RNase E that forms after long term nitrogen starvation is distinct from the transient formation of RNase E puncta (small clusters) on the inner cytoplasmic membrane during normal cell growth.

C9) The cell fractionation results quoted in Fig. 4H need to be backed up by showing the data. The issue is where is the rest of the RNase E in the *minCDE* mutant strain.

C10) Figs. 4I and EV12. In the *minCDE* mutant background, regardless of the quantification of variance in intensity, visual inspection of the micrographs suggest that rifampicin diminishes the frequency and intensity of foci at the poles of the cell.

Please comment.

C11) Paragraph starting line 301. The authors suggest that MinD is required for release/degradation of RNA from RNase E. Since RNase E levels are normal in the minCDE mutant strain, a problem with release/degradation of RNA should result in a slowdown of mRNA degradation. How do the authors reconcile the apparent destabilization of the creA mRNA in the minCDE mutant with the suggestion that MinD is required for the release/degradation of RNA?

C12) Regarding the results in Fig. EV12. The nonpermissive temperature of the rne3071 strain is 43.5{degree sign}C, but the analysis was performed at 42{degree sign}C. Are the authors sure that they have inactivated RNase E?

*A control showing accumulation of the 9S rRNA precursor of p5S rRNA needs to be performed to validate the inactivation of RNase E.

C13) Lines 735-739. Delete duplicated reference and update citation.

C14) Paragraph starting line 381. In the context of differences in cell architecture between E. coli and Caulobacter, Gray et al., 2019, Cell 177, 1632-1648 and Carpousis et al., 2022, Annu Rev Microbiol. 76, 533-552 should be mentioned and cited here.

C15) Line 402. There is no evidence that RNase E transcript release is inhibited in the minCDE mutant strain. This is an inference. Furthermore, this inference is inconsistent with an increased rate of creA mRNA degradation in the minCDE mutant strain.

C16) Paragraph starting line 414. In the context of the speculation that RNase E might have a role in translation, recent work showing that the formation of RNase E puncta on the inner cytoplasmic membrane depends on an interaction with polyribosomes should be mentioned and cited (Hamouche et al., 2021, mBio 12(5):e0193221).

C17) Line 152. Indicate that the proteomics work is presented later in the Results section.

C18) Paragraph starting line 252. The work mentioned here is at best weak evidence for an essential interaction between MinD and RNase E since it is based on the failure to construct strains. Furthermore, how do the authors reconcile the suggestion that the MinD-RNase E interaction is essential with the fact that the minCDE deletion strain is viable?

In addition, as written, it is not easy to read this paragraph. I suggest that the authors leave out the details of the failed strain constructions and list what they wanted to test. For example, if MinD is overexpressed, does it affect the localization of RNase E?

Questions

Q1) What are the r values in Fig. EV3? How were they calculated?

Q2) What are the units of the y-axis in the graph in Fig. 4A? How was the standard deviation calculated? That is, the analysis is the standard deviation of what? Pixel intensity?

Q3) With respect to Movie S1, what are we seeing? The movie needs a legend describing how this experiment was performed. In the text, the authors say the focus of RNase E at the pole of the cell is transient. What is the basis for this statement? In the movie, why are there no foci at 0 min.? Furthermore, instead of showing a single cell, a field of cells should be shown.

*If the foci form transiently, then we need to see a movie in which a foci first form and then disappear. A good way to analyze this would be to produce a kymogram as was done in the Strahl et al. 2015 reference.

Q4) Why is the error bar for WT in Fig. 4G not centered on the top of the bar?

Referee #2:

In this manuscript, the authors show that cellular MinD affects the localization of RNaseE pools in the cells and, as a result, mRNAs pool distribution. While it is currently unclear how some mRNAs would preferentially localize over others, the results suggest that in the absence of MinD, RNase E complexes accumulate at the cell pole. The authors suggest that it is attributable to a direct interaction between MinD and RNaseE, which was mapped in silico. The assumption is that MinD oscillation is a component of disrupting RNaseE polar foci - this could be further tested through analysis of MinD mutant proteins. In general, this is an interesting paper that is likely to be of broad interest. The authors are asked to address the following concerns:

Deletion of minC would also modify expression of minD due to polar effects, since the operon organization is minC-minD-minE. How is this circumvented? A western blot would confirm if MinD is still produced and at similar levels as wt.

Are RNase E foci dynamically localized in the cells (i.e., do they move around), or do they move refractory to MinD oscillation?

Fig. 4A. What percentage of minCDE- cells display polar localization of RNase E-Yfp? Is the pattern unipolar or bipolar. If oscillating MinD disperses RNase E, why would foci not be present equally at both poles?

4E. What do the bands correspond to (upper, middle, and lower)? The middle is presumably RNaseE. An RNaseE immunoblot would confirm that this middle band is RNase E and if the EdeltaA is stable. Also, was a renaturation step required for the far western?

In the bacterial two hybrid assay in Fig. 4G, MinD mutant proteins (L48R, V147R, and L194R) are less able to bind to RNE'. It is equally possible that they are degraded/unstable. How do the authors know that the variants are stable? This is also a concern for the T18-RNE'deltaA in 4F.

Fig. 4H. Fractionation of the RNaseE deltaA variant should be unaffected by deletion of minCDE. This would provide a

complementary result. Can the authors also compare a control protein, whose membrane localization is not affected by MinD?

Does MinD bind to an RNase E 568-582 peptide in vitro?

Does MinD alter the activity of RNase E in vitro? What happens to RNaseE activity and mRNA pools upon MinD overexpression? I.e., are chemoreceptor arrays further mislocalized? Is mRNA turnover reduced throughout the cell?

MinE is required for MinD oscillation. If MinD oscillation is prevented by deletion of minE, but MinD is still there, is the localization of RNase E polar? Alternatively, is a non-functional MinD variant analyzed (i.e., ATPase deficient), what is the localization of RNase E?

Line 243 - "MinD-binding binding" is unclear. Please fix the text.

Referee #3:

The study provides an exciting set of connections between polarly localized RNAs, RNase E and MinD. This study shows that the polar localization of bglG and cheA depends on the MinCDE system (Fig 1) and a functional degradosome (Fig 2). In addition, the studies show, through a combination of in-silico modeling and experiments, that identified the membrane-binding site in RNase E mediates binding to MinD (Fig 4). Overall this study presents an interesting set of connections between RNase E-MinD and polar localized RNAs that should be of broad interest to the bacterial cell biology community. However, we encourage the authors to consider the following points.

Major

1. A fascinating conclusion of the manuscript is "Polar accumulation of RNase E in Δ minCDE cells resulted in destabilization and depletion of mRNAs from the poles." However, the experiments on whether Δ minCDE strain affects RNase E activity appear limited to Figure EV7, which shows the cheA transcript half-life decreases 2-fold in the minCDE deletion. Is this because of polar localization of RNase E leads to rapid degradation of the polar RNAs? To add rigor to the author's claim, they should consider extending Figure EV7 to a set of polar RNA versus a couple of RNAs that are non-polar as controls. This would significantly bolster their studies and add a high-impact result.
2. The ways RNAs are labeled can potentially affect their subcellular localization pattern. However, the labeling scheme is buried and a bit vague. On pg 4, line 108, the authors should provide 1-2 sentences that fully describe that RNA labeling scheme. This should describe that the transcripts visualized by microscopy are chimeras with MS2 binding hairpins (specify how many), and co-expressed MS2-GFP. Making these details clear in the results text and, potentially, the figure captions will help readers interpret the observed fluorescence data.
3. The results in Fig 4I are both surprising and interesting. However, the study lacks an essential positive control: how rifampicin impacts RNase E localization in wild-type cells. A side-by-side rifampicin treatment of wild-type cells should be performed to confirm in their experimental setup that RNase E becomes diffuse across the membrane in wild-type cells. This control will add rigor to the observation in Fig 4I
4. A central unresolved question is if the cheA mRNA directly or indirectly controls the localization of the CheA signaling protein. Several questions arise from this part of the story.
 - a. Does CheA bind directly to MinCDE?
 - b. Does CheA bind directly to its mRNA? Notably, it appears CheA shows up in Fig 3B. The authors may consider demonstrating the binding of CheA to the MinCDE system or its mRNA transcript.
 - c. Does MinCDE impact the phosphorylation state of CheA? What is the relationship between CheA phosphorylation and its colocalization with the Tar receptors?
5. In Figure 2's phase images, what are the light gray foci throughout the cell body? The phase images are expected to show dark homogenous cells. Is this an overlay image of the fluorescence and phase imaging?
6. What is the proper nomenclature for labeled transcripts in Fig 2? As written, it may be misinterpreted that the transcripts are unlabeled or not modified. The visualized transcripts are chimeric transcripts containing the MS2 binding hairpin, which should be made clear to readers.
7. Figure 2 is currently lacking a quantitative analysis of images. This would allow the authors to analyze more cells in the population than the ones shown in the image below. For example, the authors may consider the analysis of foci/cell for greater than 200 cells per condition/strain. Doing this analysis would add rigor to their interpretations of images.
8. Similarly, Fig 4 is one of the critical interpretation figures of the paper. Fig 4A, amongst the 4 cells shown, only cell one shows

polar foci. To add rigor to the authors' assertion about polar localization, a quantitative analysis of images is needed in Fig 4A that categorizes cells as pole localized vs. non-pole localized for several cells (>100).

9. What is the subcellular localization pattern of RNase E in the MinD L48R, V147R, and L194R variants? This would add a line of evidence that the direct interaction between MinD-RNase E regulates RnaseE subcellular localization.

10. Figures 4I, 5C, and 5D are key figures with exciting data. However, the imaging quality is poor. Since the cells are densely packed on the image pad, it is challenging to deconvolute signals from neighboring cells. To improve the presentation of this data, the authors should properly dilute the cells on the imaging pad such that cells are not overcrowded and touching. This would make the reader's interpretation of cell images more straightforward.

Minor

1. Regarding Fig 4H, a 30% reduction in localized RNase E was observed in the DminCDE. Can you rationalize this based on the copy numbers of RNase E and MinD

2. Instead of starting the abstract with "we have recently shown," the authors should consider a more timeless way of introducing the article. Imagine reading the publication 10 years later when we're all reading.

3. On pg 11, line 323 RNase E should be rewritten as RNase E. Also, the authors should check this spelling throughout the manuscript.

4. Some font sizes in Figure 1A are very small. In addition, the fluorescence intensity profiles should have x-axis and y-axis labels.

5. In Figure EV11A, the fluorescence versus cell length should have tick marks and numerical values on each axis.

We thank the reviewers for their careful reading of the manuscript and for their different suggestions, which were raised in such a positive attitude. We feel that addressing them improved our manuscript.

Below are our point-by-point responses to the concerns of the three referees. Our answers, describing the changes, additions and clarifications that we made in the manuscript, are detailed below in blue font. The additions to the text are also highlighted in the revised manuscript by a blue font.

The revised manuscript includes new data presented in **new main figures: 4B, 4C and 6B** and **new EV figures: 3B, 5, 10, 11, 13, 17, 18, 19 and 21**.

Additionally, statistics and missing details were added to many figures, as described below.

Referee #1:

...Overall, the manuscript is logically organized and well written.

In the comments and questions, issues that need to be addressed by additional experimental work are marked with an asterisk.

Although only 4 points were marked with asterisks (highlighted in yellow below), we addressed all points and performed the required experimentation as described below.

Comments:

C1) A strain and plasmid table indicating strain background, genotypes, and key plasmid features (backbone, ori, drug resistance and constructs for expression of mRNA and protein) needs to be added as part of the requirement that the work can be reproduced by other researchers in the field.

We added the information.

C2) Regarding the results in Fig. 3, assuming that the bglG-MS2-GFP mRNA is translated, I would have expected to see more ribosomal proteins.

We do see ribosomal proteins, with some being statistically significant in the two lists (the *cheA*-binding list has been added, as required by this referee; see below). This is mentioned in lines 225-227 in the revised text. Of note, if we put the cutoff of the p value higher (proteins in both heat maps were filtered to have p value <0.01 or lower), we see more proteins in general, including additional ribosomal proteins.

Also, what are possible explanations for the pulldown of two subunits of DNA dependent RNA polymerase? Please comment.

In fact, pulldown of RNA polymerase subunits is cool, in light of the fact that BglG is a transcriptional antiterminator. We added this information to the revised text (lines 220-221).

*Most importantly, this protocol should be validated by pulling down a second mRNA. For example, cheA-MS2-GFP.

We repeated the protocol with *cheA* RNA, as suggested by the reviewer. The results are presented in **new Fig EV10** and are described in lines 221-223. The mass spectrometry proteomics data have been deposited and the details are in the Method section.

C3) Fig. 4B and text. The conclusion that minCDE is responsible for mobility of RNase E is an overinterpretation of the FRAP results. If RNase E is detached from the membrane and trapped in the focus, then reduced diffusion is an indirect effect of the minCDE knockout. It is an error to suggest that minCDE actively promotes RNase E diffusion in the wild type cell.

Our results show that approx. two thirds of the RNase E molecules are still membrane-associated in Δ *minCDE* cells, some near the pole and others along the cell's circumference. We forgot to mention that in the FRAP we photobleached membrane foci, not only at the pole, and the recovery of these membrane foci was monitored. In the revised text, we mention this, but also suggest that the effect might be related to the effect of the Min system on the association of RNase E with the membrane, as proposed by the referee (lines 339-346). Of note, we moved this paragraph to the section that deals with the effect of MinD on RNase E interaction with the membrane.

C4) The data in Fig. EV7 and text. The conclusion that *cheA* mRNA is selectively destabilized in the *minCDE* mutant is an over interpretation of the experimental result.

*Additional stability measurements are required including at least one more mRNA localized to the cell pole (*bglB*?) as well as control mRNAs that are not localized to the cell pole.

As requested by the referee, we performed additional stability measurements. The results with three mRNAs that are predicted to be enriched at the poles and three not predicted to be enriched at the poles are presented in **new Fig. 4C** and **EV13**, and are described in lines 255-262. This was a very good suggestion, since the difference between the two groups seems clear. (A comment: we did not monitor *bglB*, because the *bgl* operon is cryptic, so we would have to use a different strain and add β -glucosides to follow *bglB* expression, and that would be an experiment at different conditions than those used for the other mRNAs)

C5) Line 900. The title is mislabeled. This is the legend for Fig. 5

Corrected.

C6) Labelling of Far Western blot in Fig. 4E is confusing. The heading 'Blot reacted with:' is not necessary since it is the same for both lanes. Also 'In gel lysate overexpressing:' is unnecessary. The panel could be simply labelled 'Far Western Blot'.

Corrected as the reviewer suggested.

C7) There is a discrepancy in the labelling of Panels C and G in Fig. 4. Panel C indicates 3 leucine residues: L48, L147 and L198. Panel G indicates a V147R mutation. The text also mentions V147. Considering the model in Panel C, it looks like V147 is mislabeled. The arrow points to a leucine residue in segment-A of RNase E. I suspect that V147 is a MinD residue slightly to the left of the leucine in segment-A of RNase E.

The referee is right. The model and labeling of V147 was corrected. We apologize for the mistake.

C8) Line 304. The citation of McQuail et al. 2022 here is misleading. The large focus of RNase E that forms after long term nitrogen starvation is distinct from the transient formation of RNase E puncta (small clusters) on the inner cytoplasmic membrane during normal cell growth.

The citation of McQuail et al. 2022 was omitted from this paragraph. The issue of large polar condensates is only mentioned now in the Discussion, where we clearly distinguish between the appearance and the localization patterns of RNase E in different growth conditions and bacteria (lines 440-443).

C9) The cell fractionation results quoted in Fig. 4H need to be backed up by showing the data. The issue is where is the rest of the RNase E in the minCDE mutant strain.

We repeated the fractionation experiment. The results in **new Fig. 6B** show a decrease in RNase E in the membrane of $\Delta minCDE$ cells and a reciprocal increase in the cytoplasm, compared to wild-type cells. Notably, the total amount of protein does not change between the two strains.

C10) Figs. 4I and EV12. In the minCDE mutant background, regardless of the quantification of variance in intensity, visual inspection of the micrographs suggest that rifampicin diminishes the frequency and intensity of foci at the poles of the cell. Please comment.

Unfortunately, this was a false impression due to overlap between the cells. We replaced the images with images that enable better visualization of separate cells (see **Fig. 6D**, which contains the data and analysis for both $\Delta minCDE$ and wild type cells side-by-side; previously presented in Figs 4I and EV12).

C11) Paragraph starting line 301. The authors suggest that MinD is required for release/degradation of RNA from RNase E. Since RNase E levels are normal in the minCDE mutant strain, a problem with release/degradation of RNA should result in a slowdown of mRNA degradation. How do the authors reconcile the apparent destabilization of the *cheA* mRNA in the minCDE mutant with the suggestion that MinD is required for the release/degradation of RNA?

I hope that in the revised text this issue is better explained. We explain that MinD is required for release/degradation from/by RNase E of mainly or only polar mRNAs since in wild-type cells, as opposed to $\Delta minCDE$ cells, RNase E is almost depleted from the poles. The destabilization of the *cheA* and additional polar mRNAs in $\Delta minCDE$ is due to the movement of RNase E to the poles.

C12) Regarding the results in Fig. EV12. The nonpermissive temperature of the rne3071 strain is **43.5**{degree sign}C, but the analysis was performed at **42**{degree sign}C. Are the authors sure that they have inactivated RNase E? *A control showing accumulation of the 9S rRNA precursor of p5S rRNA needs to be performed to validate the inactivation of RNase E.

In fact, the non-permissive temperature is 43 (McDowall et al., 1993). We preformed qPCR for the 9S rRNA precursor of p5S rRNA in 30°C, 42°C and 43°C, as suggested by the referee. The results in **new Fig. EV5** show accumulation of the precursor in both 42°C and 43°C (restrictive temperature), as opposed to 30°C (the permissive temperature).

C13) Lines 735-739. Delete duplicated reference and update citation.

Both corrected.

C14) Paragraph starting line 381. In the context of differences in cell architecture between E. coli and Caulobacter, Gray et al., 2019, Cell 177, 1632-1648 and Carpousis et al., 2022, Annu Rev Microbiol. 76, 533-552 should be mentioned and cited here.

These references were added. Thanks.

C15) Line 402. There is no evidence that RNase E transcript release is inhibited in the *minCDE* mutant strain. This is an inference. Furthermore, this inference is inconsistent with an increased rate of *cheA* mRNA degradation in the *minCDE* mutant strain.

I hope that our response to C11 answers this point.

C16) Paragraph starting line 414. In the context of the speculation that RNase E might have a role in translation, recent work showing that the formation of RNase E puncta on the inner cytoplasmic membrane depends on an interaction with polyribosomes should be mentioned and cited (Hamouche et al., 2021, mBio 12(5):e0193221).

The reference was added. A great contribution that supports our speculation. Thanks!

C17) Line 152. Indicate that the proteomics work is presented later in the Results section.

In the revised text, this is not mentioned in this location anymore, only when reported.

C18) Paragraph starting line 252. The work mentioned here is at best weak evidence for an essential interaction between MinD and RNase E since it is based on the failure to construct strains. Furthermore, how do the authors reconcile the suggestion that the MinD-RNase E interaction is essential with the fact that the *minCDE* deletion strain is viable? In addition, as written, it is not easy to read this paragraph. I suggest that the authors leave out the details of the failed strain constructions and list what they wanted to test. For example, if MinD is overexpressed, does it affect the localization of RNase E?

We agree with the reviewer that we went too far in drawing conclusions from negative results. We decided to include them in the first place, thinking that the reader would wonder why this has not been done. We shortened this paragraph to include only the essential details, without drawing conclusions (lines 279-283).

We also tested the effect of MinD overexpression on RNase E localization as suggested by the reviewer. The results are shown in **new Fig. EV11**.

Questions

Q1) What are the *r* values in Fig. EV3? How were they calculated?

"*r*" is the Pearson's correlation coefficient that was calculated by NIS-elements. This data was added to the legend of Fig. EV7.

Q2) What are the units of the *y*-axis in the graph in Fig. 4A? How was the standard deviation calculated? That is, the analysis is the standard deviation of what? Pixel intensity?

Y-axis labeling was added to Fig. 4A graph. *Y*-axis is the standard deviation. The standard deviation was calculated for the fluorescence intensity inside the cell using NIS-elements. The standard deviation of fluorescence intensity of a cell is indirectly proportional to its uniform distribution. Therefore, in *ΔminCDE*, the polar clusters results in higher standard deviation compared to wild type where there are no polar clusters. This information was added to the figure legend.

Q3) With respect to Movie S1, what are we seeing? The movie needs a legend describing how this experiment was performed. In the text, the authors say the focus of RNase E at the pole of the cell is transient. What is the basis for this statement? In the movie, why are there no foci at 0 min.? Furthermore, instead of showing a single cell, a field of cells should be shown.

*If the foci form transiently, then we need to see a movie in which a foci first form and then disappear. A good way to analyze this would be to produce a kymogram as was done in the Strahl et al. 2015 reference.

We extended EV movie 1 and it is now showing the appearance and dispersal of clusters over time. We also added a Kymograph showing the transient nature of the cluster, as suggested by the referee (new Fig. 4B). We focused on one cell because not all cells have such a foci that is easy to follow (the percentage is given in the text).

Q4) Why is the error bar for WT in Fig. 4G not centered on the top of the bar?

Corrected.

Referee #2:

... In general, this is an interesting paper that is likely to be of broad interest. The authors are asked to address the following concerns:

1. Deletion of *minC* would also modify expression of *minD* due to polar effects, since the operon organization is *minC*-*minD*-*minE*. How is the circumvented? A western blot would confirm if *MinD* is still produced and at similar levels as wt.

To test if *MinD* is still produced in *minC* deletion strain, we preformed Western blot analysis, as suggested by the referee, using anti-*MinD* antiserum. The result in new Fig. EV3B implies that *minC* deletion does not have a polar effect on the expression of the *minD* gene.

2. Are RNase E foci dynamically localized in the cells (i.e., do they move around), or do they move refractory to *MinD* oscillation?

To address this point, we added new EV movie 2, which shows that RNase E is dynamic and that its movement does not correlate with *MinD* oscillation.

3. Fig. 4A. What percentage of *minCDE*- cells display polar localization of RNase E-Yfp? Is the pattern unipolar or bipolar. If oscillating *MinD* disperses RNase E, why would foci not be present equally at both poles?

Twelve percent (65 out of 542) of the cells present a polar cluster. This information was added to the figure legend. The pattern is unipolar, since the poles of *E. coli* cells are not identical, one being old and the other new. In fact, studying polar localization of proteins and RNAs for many years now, I can say that, when expressed from the chromosome, both proteins and RNAs are always detected only in one pole of *E. coli* cells. Expression from a plasmid results in localization to both poles, showing their capacity.

4. Fig. 4E. What do the bands correspond to (upper, middle, and lower)? The middle is presumably RnaseE. An RNaseE immunoblot would confirm that this middle band is RNase E and if the Δ EdeltaA is stable. Also, was a renaturation step required for the far western?

The RNase E is indeed the middle one, now labeled with an arrowhead. This is evident from the Coomassie-stained half gel that is now presented next to the full membrane blotted from the second half gel, which was used for the Far-Western, in Fig. EV16. Regarding the renaturation, proteins were not boiled before conducting the SDS-PAGE. Renaturation occurs during the semi-dry transfer.

5. In the bacterial two hybrid assay in Fig. 4G, *MinD* mutant proteins (L48R, V147R, and L194R) are less able to bind to RNE'. It is equally possible that they are degraded/unstable. How do the authors know that the variants are stable? This is also a concern for the T18-RNE' Δ deltaA in 4F.

In response to this point, we compared the steady-state level of the *MinD* mutants to that of the wild type by Western blot analysis, using anti-*MinD* antiserum. The result in new Fig. EV17 confirms that the level of mutant proteins is comparable to that of the wild type. The stability of the RNase E Δ A variant was examined before and shown to be even slightly higher than that of the wild type (see

Khemici et al., Mol micro, 2008). All this information was added to the revised text (lines 293-295 and lines 309-314).

6. Fig. 4H. Fractionation of the RNaseE deltaA variant should be unaffected by deletion of minCDE. This would provide a complementary result.

Segment A is what brings RNase E to the membrane. Hence, the RNase E delta A variant is not in the membrane fraction, regardless of whether the strain carries the *minCDE* operon or not.

Can the authors also compare a control protein, whose membrane localization is not affected by MinD?

This is not a standard control for this type of assay, since the conclusions are based on the difference between the phenotypes presented by different strains (wild type and delta *minCDE* in this case). In addition, MinD is a central protein and no one can say with certainty which proteins are not affected by it.

7. Does MinD bind to an RNase E 568-582 peptide in vitro?

I am not sure what binding to a purified peptide, which is out of context and most probably does not fold correctly, might prove. In addition, how can we purify it without tagging it?

8. Does MinD alter the activity of RNase E in vitro?

The fact that RNase E is a membrane protein and the requirement for the entire degradosome complex for degradation are probably the reasons why RNase E has not been tested *in vitro* ever.

What happens to RNaseE activity and mRNA pools upon MinD overexpression? I.e., are chemoreceptor arrays further mislocalized? Is mRNA turnover reduced throughout the cell?

The effect of MinD overexpression on CheA and TaR localization was tested, as suggested by the referee. The results, presented in **new Fig. EV21**, show that overexpression of MinD causes mislocalization of the CheA protein, similar to the effect of the *minCDE* deletion, whereas, the chemoreceptor localization is not affected. This is a nice addition and we thank the reviewer for that.

9. MinE is required for MinD oscillation. If MinD oscillation is prevented by deletion of minE, but MinD is still there, is the localization of RNase E polar?

The *minE* gene alone cannot be deleted, because MinC and MinD will prevent Z ring formation all along the membrane, causing lethal filamentation (de Boer et al., *Cell* 56:641, 1989). To delete *minE*, one would need to overexpress FtsZ. This is explained in lines 165-167.

Alternatively, is a non-functional MinD variant is analyzed (i.e., ATPase deficient), what is the localization of RNase E?

In response to this question, we tested RNase E localization in an ATPase-deficient MinD mutant (K16Q). The results, presented in **new Fig. EV19**, show that no change in RNase E puncta distribution along the membrane in cells expressing the non-functional ATPase-deficient MinD mutant.

10. Line 243 - "MinD-binding binding" is unclear. Please fix the text.

Corrected

Referee #3:

...Overall this study presents an interesting set of connections between RNase E-MinD and polar localized RNAs that should be of broad interest to the bacterial cell biology community. However, we encourage the authors to consider the following points.

Major

1. A fascinating conclusion of the manuscript is "Polar accumulation of RNase E in Δ minCDE cells resulted in destabilization and depletion of mRNAs from the poles." However, the experiments on whether Δ minCDE strain affects RNase E activity appear limited to Figure EV7, which shows the *cheA* transcript half-life decreases 2-fold in the minCDE deletion. Is this because of polar localization of RNase E leads to rapid degradation of the polar RNAs? To add rigor to the author's claim, they should consider extending Figure EV7 to a set of polar RNA versus a couple of RNAs that are non-polar as controls. This would significantly bolster their studies and add a high-impact result.

As suggested by the referee, we performed additional stability measurements. The results with three mRNAs that are predicted to be enriched at the poles and three not predicted to be enriched at the poles are presented in **new Fig. 4C** and **EV13** and described in lines 257-260. This was a very good suggestion and I hope that the referee agrees with us that the difference between the two groups are significant.

2. The ways RNAs are labeled can potentially affect their subcellular localization pattern. However, the labeling scheme is buried and a bit vague. On pg 4, line 108, the authors should provide 1-2 sentences that fully describe that RNA labeling scheme. This should describe that the transcripts visualized by microscopy are chimeras of MS2 binding hairpins (specify how many), and co-expressed MS2-GFP. Making these details clear in the results text and, potentially, the figure captions will help readers interpret the observed fluorescence data.

We apologize for not including this information in the first place. We now added it (lines 119-125).

3. The results in Fig 4I are both surprising and interesting. However, the study lacks an essential positive control: how rifampicin impacts RNase E localization in wild-type cells. A side-by-side rifampicin treatment of wild-type cells should be performed to confirm in their experimental setup that RNase E becomes diffuse across the membrane in wild-type cells. This control will add rigor to the observation in Fig 4I.

We did perform the experiment also with the wild type. In fact, the result was in an EV figure and we forgot to mention it in the text. It has now been moved to be side-by-side with the main figure, as suggested by the referee (**new Fig. 6D**).

4. A central unresolved question is if the *cheA* mRNA directly or indirectly controls the localization of the CheA signaling protein. Several questions arise from this part of the story.

a. Does CheA bind directly to MinCDE?

b. Does CheA bind directly to its mRNA? Notably, it appears CheA shows up in Fig 3B. The authors may consider demonstrating the binding of CheA to the MinCDE system or its mRNA transcript.

c. Does MinCDE impact the phosphorylation state of CheA? What is the relationship between CheA phosphorylation and its colocalization with the Tar receptors?

a. CheA is prevalent at the poles. Being fascinated by chemotaxis, I agree with the referee that these are interesting questions. However, for this study, why is CheA more interesting than the other proteins that came down with the *bglG* and *cheA* transcripts? b. We do not rule out involvement of other proteins in localizing polar transcripts. c. Our manuscript does not deal with how localization of RNAs determines protein localization, although it is most likely by localized translation (I hope to address this question one day). Hence, testing the relationship between CheA phosphorylation and its co-localization with the Tar receptors does not seem much related to RNA localization (with my background on protein phosphorylation, I sympathize with the referee's curiosity, but these questions should be addressed in the future).

5. In Figure 2's phase images, what are the light gray foci throughout the cell body? The phase images are expected to show dark homogenous cells. Is this an overlay image of the fluorescence and phase imaging?

We often observe such an appearance in phase, especially when using of PBS for bacterial resuspension or in the agar pad (as if big clusters are somewhat transparent in phase).

6. What is the proper nomenclature for labeled transcripts in Fig 2? As written, it may be misinterpreted that the transcripts are unlabeled or not modified. The visualized transcripts are chimeric transcripts containing the MS2 binding hairpin, which should be made clear to readers.

We added an explanation on the MS2 system and the chimeras in response to point 2. The names of the MS2 chimeric transcript were changed and they are now as in Nevo-dinur et al., *Science* 2011, in all figures.

7. Figure 2 is currently lacking a quantitative analysis of images. This would allow the authors to analyze more cells in the population than the ones shown in the image below. For example, the authors may consider the analysis of foci/cell for greater than 200 cells per condition/strain. Doing this analysis would add rigor to their interpretations of images.

We preformed the analysis as suggested by the referee (for $n > 200$) (see **new Fig. EV6**).

8. Similarly, Fig 4 is one of the critical interpretation figures of the paper. Fig 4A, amongst the 4 cells shown, only cell one shows polar foci. To add rigor to the authors' assertion about polar localization, a quantitative analysis of images is needed in Fig 4A that categorizes cells as pole localized vs. non-pole localized for several cells (>100).

We preformed the analysis as suggested by the referee. 12% (65 out of 542) of the cells present a polar cluster. This data was added to the figure legend.

9. What is the subcellular localization pattern of RNase E in the MinD L48R, V147R, and L194R variants? This would add a line of evidence that the direct interaction between MinD-RNase E regulates RNase E subcellular localization.

In response to this question, we compared the localization patterns of the three MinD variants to that of wild type (see **new Fig. EV18**). However, this should be considered together with the results reported in **new EV Movie 2**, which shows that MinD pole-to pole oscillation and RNase E dynamics in the membrane are not correlated, as well as the result in **new Fig. EV19**, which shows lack of effect of an ATPase-deficient MinD mutant on RNase distribution.

10. Figures 4I, 5C, and 5D are key figures with exciting data. However, the imaging quality is poor. Since the cells are densely packed on the image pad, it is challenging to deconvolute signals from neighboring cells. To improve the presentation of this data, the authors should properly dilute the cells on the imaging pad such that cells are not overcrowded and touching. This would make the reader's interpretation of cell images more straightforward.

We repeated the experiment as suggested by the referee with cells more diluted on the imaging pad (see **Fig. 7C and 7D**) or cropped a smaller field (see **Fig. 6D**). Of note, since cells borders are easily seen in the new images that are presented, phase images are no longer presented. This should enable the presentation of bigger images in the final figure.

Minor

1. Regarding Fig 4H, a 30% reduction in localized RNase E was observed in the DminCDE. Can you rationalize this based on the copy numbers of RNase E and MinD?

The estimates regarding the copy numbers of RNase E and MinD, derived from genome-wide ribosome profiling, indicate that the number of RNase E molecules is approx. 2-fold lower than that of MinD. This information and the rational it provides to the extent of MinD influence on RNase E were added to the revised text (lines 334-338).

2. Instead of starting the abstract with "we have recently shown," the authors should consider a more timeless way of introducing the article. Imagine reading the publication 10 years later when we're all reading.

We deleted the word recently (with a smile, after imagining what the referee suggested).

3. On pg 11, line 323 RNase E should be rewritten as RNase E. Also, the authors should check this spelling throughout the manuscript.

Corrected.

4. Some font sizes in Figure 1A are very small. In addition, the fluorescence intensity profiles should have x-axis and y-axis labels.

The figure was formatted according to the reviewer comment and the axes were labeled.

5. In Figure EV11A, the fluorescence versus cell length should have tick marks and numerical values on each axis.

The Y-axis was corrected and its definition is in the figure legend (no space for that in the figure itself). The X-axis presents the cell length as indicated. Since cells differ in their length, we plotted the fluorescence intensity profiles along the long cell axis after normalizing to cell length. To clarify it to the readers, we added this information to the figure legends throughout the manuscript.

I hope that our response to the comments raised by the referees is satisfactory and that the manuscript is now suitable for publication in *EMBO J*.

I look forward to hearing from you. Best wishes,

Sincerely,

Orna.

Prof. Orna Amster-Choder
Department of Molecular Biology
The Hebrew University Medical School
P.O.B. 12272, Jerusalem 91120, ISRAEL
Fax: 972 2 675 7910; Tel: 972 2 675 8460
e mail: ornaam@ekmd.huji.ac.il

Dear Orna,

Thank you for submitting your revised manuscript to The EMBO Journal. Your manuscript has now been seen by two of the original reviewers, and you can find their comments below.

As you can see, both reviewers appreciate the revision work, while reviewer #3 finds that further data re-analysis and softening of some statements would be needed. Furthermore, referee #1 provides a suggestion for addressing the more challenging point 7 by referee #2, who was not able to re-assess the manuscript. While, as discussed, I understand the difficulties in performing these experiments, please take a look whether their proposed approach would be feasible in your team, although this will not be absolutely required for the acceptance.

Based on this input, I would like to invite you to address the remaining comments by reviewer #3 and the following editorial issues:

1. We are missing the ORCID iD for the co-corresponding author Shanmugapriya Kannaiah. In order to link the ORCID iD to the account in our manuscript tracking system, the author in question has to do the following:
 - Click the 'Modify Profile' link at the bottom of your homepage in our system.
 - On the next page you will see a box halfway down the page titled ORCID*. Below this box is red text reading 'To Register/Link to ORCID, click here'. Please follow that link: you will be taken to ORCID where you can log in to your account (or create an account if you don't have one)
 - You will then be asked to authorise Wiley to access your ORCID information. Once you have approved the linking, you will be brought back to our manuscript system.Unfortunately, we cannot do this linking on the author's behalf for security reasons.
2. Please upload the main and EV figures as individual production quality figure files in the .eps, .tif, or .jpg format (one file per figure).
3. We can accommodate up to five EV figures. The remaining figures and their legends should be renamed Appendix Figure S1 etc. and compiled in one Appendix PDF file, with their legends and a brief, numbered table of contents added. Please upload the Appendix file as an expanded view file type labelled "Appendix". Please remove the legends of Appendix Figures from the manuscript text files.
4. Please remove movie legends from the manuscript and zip with each corresponding movie file. Please correct the nomenclature to Movie EV1 and EV2 in the legends, callouts and movies themselves.
5. Please upload tables EV1-3 as individual, editable files. Please add the references to the main manuscript references.
6. Please upload the Reagents and Tools table as a separate table. When using this format, please add lanes referring to the information on the used strains, oligonucleotides and plasmids in EV tables (please see the attached template).
7. Please add the Data Availability section to the manuscript that should contain the information on deposition and access to the mass spectrometry dataset. Please also include a resolvable link to the dataset and remove the referee token. Further information on this section and its format can be found here:
<https://www.embopress.org/page/journal/14602075/authorguide#dataavailability>.
8. Our data editors have flagged the following issues in figure legends that need correcting:
 - The legend of figure 5E refers to panel F ("as in F"); however, no figure panel F is provided in the corresponding figure.
 - Although 'n' is provided, please describe the nature of the replicates (e.g., biological or technical) in the legends of figures 1a-b; EV2a-c; EV3a; EV4
 - The error bars are not defined in the legend of figures 4a; 6a, c
 - The number and nature of replicates (n) is missing in the legend of figures 6a, c; EV5.
9. Please rename "Conflict of interest" section into "Disclosure and competing interests statement" (further info:
<https://www.embopress.org/page/journal/14602075/authorguide#conflictofinterest>).
10. Papers published in The EMBO Journal are accompanied online by a 'Synopsis' to enhance discoverability of the manuscript. Please submit synopsis text that consists of A) a short (1-2 sentences) summary of the findings and their significance, B) 3-4 bullet points highlighting key results and C) a synopsis image that is 550x300-600 pixels large (width x height, jpeg or png format). You can either show a model or key data in the synopsis image. Please note that the image size is rather small, and that text needs to be readable at the final size.

Please feel free to contact me if you have any questions regarding this final editorial revision. Please use the link below to upload the revised files.

Thank you for the opportunity to consider your work for publication, and I look forward to receiving your revised manuscript.

With best wishes,

leva

We realize that it is difficult to revise to a specific deadline. In the interest of protecting the conceptual advance provided by the work, we recommend a revision within 3 months (2nd Jan 2024). Please discuss the revision progress ahead of this time with the editor if you require more time to complete the revisions.

Referee #1:

In this revised manuscript the authors present experimental work on the search for factors involved in the localization of mRNA to the poles of the rod-shaped *E. coli* cell. Two approaches were used. In the first, the polar localization of bglB and cheA mRNA was used to screen candidate factors such as cell division proteins. In the second, a proteomics approach was used to identify proteins that co-purified with bglG-MS2-GFP mRNA. The cell division protein MinD and the RNA degradosome component RNase E were identified as factors that affected the localization of bglG and cheA mRNA. Using a structure-based in silico approach, the authors predicted an interaction between the membrane attachment site of RNase E and MinD. The interaction was validated experimentally by Far Western blotting and two hybrid analyses including mutation of residues in MinD that were predicted to be necessary for the interaction with RNase E. Further experimental work investigated the effect of the deletion of the minCDE operon on the localization of RNase E, the cheA mRNA, and the TAR and EI proteins. Overall, the manuscript is logically organized and well written.

The authors have revised the manuscript along the line suggested by the referees including a substantial amount of new experimental work. With one exception, I believe that the authors have adequately responded to all the referees' queries. Regarding Referee #2, point 7. There is no reason to believe that a synthetic peptide corresponding to the RNase E membrane attachment site would not form an amphipathic alpha-helix. Indeed, binding to MinD could stabilize the alpha-helix (see Khemici 2008). Interaction of the peptide with MinD could be measured by ITC. Direct evidence for an interaction would significantly reinforce a major conclusion of this work.

Referee #3:

The authors have diligently considered the reviewer's comments and concerns. Several new experiments were added that I list below

- stability measurements of three mRNAs that are predicted to be enriched at the poles and three not predicted to be enriched at the poles are presented in new Fig. 4C and EV13
- A new study that enriched proteins via a cheA mRNA pulldown (Fig EV10),
- an improved cell fractionation experiment in Fig 6B,
- a new analysis of the RNase E temperature-sensitive mutant at 42 and 43 C to increase rigor in EV5,
- new Fig. EV3B implies that minC deletion does not have a polar effect on the expression of the minD gene.
- new EV movie 2 shows that RNase E is dynamic and that its movement does not correlate with MinD oscillation.
- new Fig. EV21 shows that overexpression of MinD causes mislocalization of the CheA protein,

I have a few remaining comments the authors should consider. These comments should only require minor re-analysis or adjustments to the text. No new experiments are discussed by the reviewer.

specific major concerns essential to be addressed to support the conclusions

1. Decay curves for gadE and motA in Fig EV13 require reanalysis. Many of the points do not fit well with these curves. This causes the authors to overestimate the half-lives of these RNAs. For example, at 3 min, the signal of gadE in the minCDE deletion is <20%. However, the half-life is listed as 3.7 min. According to the data, gadE half-life should be well below 3 min. A similar reanalysis should be done for motA.

2. Related to this point, error bars and significance analysis should be performed to understand the significance of the decay rate difference in Fig 4C.

3. The abstract makes one statement I'm not sure is well supported by the data: "Intriguingly, not only does MinD affect RNase E interaction with the membrane, but it also affects its mode of action and dynamics." The authors provided some experiments that may counter this conclusion (see below). Could the authors clarify this conclusion in light of their new results?

- New EV movie 2 shows that RNase E is dynamic and that its movement does not correlate with MinD oscillation.
- Figure EV18. MinD point mutations do not cause changes in RNase E localization

4. Line 347-349: Regarding Reviewer 1's concerns, I might suggest softening the interpretation. Add the words "directly or indirectly" when stating the "Min System is involved" directly or indirectly

"These results suggest that the Min system is involved in RNase E dynamics/diffusion in the membrane, although the lack of recovery in Δ minCDE cells might also be related to the effect of the Min system on the association of RNase E with the membrane."

- minor concerns that should be addressed

5. Since Fig EV11 appears to be a negative result, do the authors have any evidence that MinD overexpression occurred (e.g., western blot or in-cell fluorescence from a MinD FP fusion)? Is the mild cell filamentation what you expect from a MinD overexpression based on the author's experience or past publications? It's possible that MinD expression levels are not as high as would be expected.

any additional non-essential suggestions for improving the study (which will be at the author's/editor's discretion)

6. The authors chose not to make a great suggestion from Reviewer 2: "Can the authors also compare a control protein whose membrane localization is not affected by MinD?" This type of control would reveal if MinD has a specific effect on RNase E or a non-specific effect on all membrane-bound proteins. I agree it's not a canonical experiment, but it's a killer experiment in which either result would have been interesting.

Editorial issues:

All corrected, added, changes, provided, etc. following the instructions. Of note, some of the results in the EV Figures were transferred to the Main Figures and some to Appendix Figures.

Referee #1:

Regarding Referee #2, point 7. There is no reason to believe that a synthetic peptide corresponding to the RNase E membrane attachment site would not form an amphipathic alpha-helix. Indeed, binding to MinD could stabilize the alpha-helix (see Khemici 2008). Interaction of the peptide with MinD could be measured by ITC. Direct evidence for an interaction would significantly reinforce a major conclusion of this work.

Although you wrote that is not an absolute requirement for acceptance of the manuscript, we considered it very seriously the suggestion of reviewer #1, who acted on behalf of reviewer #2. After a long discussion with the expert on ITC in our faculty, we decided not to perform it. The decision is not so much because of the difficulties in performing it, but rather because the results of this *in vitro* analysis will be dubious, will not contribute much to the manuscript, and will be significantly less valuable than our *in vivo* results. Without going too much into details, most ITC experiments are conducted to study the interaction between two soluble cytosolic proteins. As the reviewer suggested, a synthetic peptide corresponding to the RNase E membrane attachment site might form an amphipathic alpha helix (or not), but, due to the intimate interaction of this peptide with the membrane, we will have to carry out preliminary tests to determine the solutes in which the interaction with the MinD protein should be tested. In such a case, which peptides shall we use as a positive control? There might be many reasons for getting false negative (or positive) results. If we see an interaction with the wild type peptide and with the mutants, which impair the interaction *in vivo*, how shall we regard the results? I could go on, but I believe that the most important argument is that we provided proofs for the interaction *in vivo* (and the mutants predicted to impair the interaction that impaired it). Had we have *in vitro* proofs for the interaction, it would be recommended to obtain *in vivo* proofs, but the opposite does not seem to justify the time it should take (to synthesize the wild type and mutant peptides, to calibrate the conditions, etc.) nor the effort.

Referee #3:

I have a few remaining comments the authors should consider. These comments should only require minor re-analysis or adjustments to the text. No new experiments are discussed by the reviewer. Specific major concerns essential to be addressed to support the conclusions.

1. Decay curved for *gadE* and *motA* in Fig EV13 require reanalysis. Many of the points do not fit well with these curves. This causes the authors to overestimate the half-lives of these RNAs. For example, at 3 min, the signal of *gadE* in the $\Delta minCDE$ deletion is <20%. However, the half-life is listed as 3.7 min. According to the data, *gadE* half-life should be well below 3 min. A similar reanalysis should be done for *motA*.

The lines in this figure were generated by "smoothed curve fit" (Excel), as mentioned in the legend and detailed (including the equations) in the Methods section. This is commonly done to get the slope to calculate half-lives. Unfortunately, the description in the Methods section did not clearly refer to smoothed curve fitting, and this has been corrected now. Notably, most curves have good R-squared values of ~0.9 or above (the R^2 indicates how good the data points fit with the generated decay curve). The only one with low R^2 (0.248) is *gadE*. The one obtained for *motA* ($R^2=0.767$) is not quite good. What is important in this case is that the non-smoothed curves in both cases (*gadE* and *motA*), if drawn, would show the same relationship between the half-lives of these mRNAs in wild type and $\Delta minCDE$ cells, as the smoothed curves show. That is, *gadE* and *motA*, which are enriched at the poles, would still show selective destabilization in $\Delta minCDE$ cells.

2. Related to this point, error bars and significance analysis should be performed to understand the significance of the decay rate difference in Fig 4C.

Done.

3. The abstract makes one statement I'm not sure is well supported by the data: "Intriguingly, not only does MinD affect RNase E interaction with the membrane, but it also affects its mode of action and dynamics." The authors provided some experiments that may counter this conclusion (see below). Could the authors clarify this conclusion in light of their new results?

- New EV movie 2 shows that RNase E is dynamic and that its movement does not correlate with MinD oscillation.
- Figure EV18. MinD point mutations do not cause changes in RNase E localization

Microscopy and cell fractionation data clearly show that MinD affects recruitment of RNase E to the membrane, as well as localization to the poles. However, the reviewer question is related to the effect of MinD on RNase E dynamics. Indeed, and this is emphasized in the manuscript, MinD pole-to pole oscillation and RNase E dynamics in the membrane are not correlated, as shown in EV Movie 2 and Fig. EV18. Still, we clearly show that, despite the lack of correlation in their dynamics, the two proteins interact (!) and hence may affect each other dynamics and activity. We provide evidence that indeed MinD affects RNase E dynamics (see FRAP results in $\Delta minCDE$ cells, as well as RNase E clusters distribution along the membrane in $\Delta minCDE$ cells after transcription inhibition). Hence, despite the difference in the type of movement, MinD affects RNase E dynamics. In response to this question, we revised the text to further emphasize that **(line 369)**.

4. Line 347-349: Regarding Reviewer 1's concerns, I might suggest softening the interpretation. Add the words "directly or indirectly" when stating the "Min System is involved" directly or indirectly...

We added "directly or indirectly", as suggested by the referee **(line 344 in the revised text)**.

- minor concerns that should be addressed

5. Since Fig EV11 appears to be a negative result, do the authors have any evidence that MinD overexpression occurred (e.g., western blot or in-cell fluorescence from a MinD FP fusion)? Is the mild cell filamentation what you expect from a MinD overexpression based on the author's experience or past publications? It's possible that MinD expression levels are not as high as would be expected. any additional non-essential suggestions for improving the study (which will be at the author's/editor's discretion)

As requested by the reviewer, we provide evidence that MinD overexpression indeed occurred by comparing the fluorescence intensity in cells uninduced and induced for MinD overexpression (see new Fig. EV5 B).

6. The authors chose not to make a great suggestion from Reviewer 2: "Can the authors also compare a control protein whose membrane localization is not affected by MinD?" This type of control would reveal if MinD has a specific effect on RNase E or a non-specific effect on all membrane-bound proteins. I agree it's not a canonical experiment, but it's a killer experiment in which either result would have been interesting.

I am afraid that I am less enthusiastic about this suggestion (raised reviewer #2 in the previous round). The conclusions in this type of assay are customarily based on the difference between phenotypes presented by different strains (wild type and delta *minCDE*, in this case). This idea could be tried for a less central protein. However, MinD is a very central protein and there is no way to say with certainty which proteins are not affected by it.

I hope that our manuscript is now suitable for publication in *EMBO J*.

I look forward to hearing from you.

Sincerely

Orna.

Prof. Orna Amster-Choder
Department of Molecular Biology
The Hebrew University Medical School
P.O.B. 12272, Jerusalem 91120, ISRAEL
Fax: 972 2 675 7910; Tel: 972 2 675 8460
e mail: ornaam@ekmd.huji.ac.il

Dear Orna,

Thank you for addressing the final issues. I am now pleased to inform you that your manuscript has been accepted for publication.

Before we forward your manuscript to our publishers, there are a couple of points that I still need your input on:

- 1) In the Data Availability section, we would need a resolvable link to the dataset. In the current version, referee access was provided, which I have removed.
- 2) I would like to propose a few minor changes in the article abstract and synopsis. I have also written a short blurb that will accompany the title of your manuscript in our online table of contents. Please take a look at the text below and in the attached manuscript text file and let me know if any corrections are necessary.

Blurb:

The polar transcriptome of *E. coli* is regulated by MinD-dependent inhibition of RNase E recruitment to the cell poles.

Synopsis:

A subset of *E. coli* transcripts localizes to the cell poles via an unknown mechanism. Here, the interaction of MinD and RNase E is shown to prevent site-specific degradation of polar mRNAs, thus favoring their enrichment in the cell poles.

- Candidate approach and proteomic analysis identify MinD and RNase E as regulators of polar mRNA localization.
- MinD interacts with a short peptide within the membrane-targeting sequence of RNase E.
- In the absence of MinD, RNase E transiently accumulates at cell poles.
- Polar accumulation of RNase E in Δ minCDE cells destabilizes polar mRNAs, resulting in their elimination from the poles, and may affect localization of the encoded proteins.

You may qualify for financial assistance for your publication charges - either via a Springer Nature fully open access agreement or an EMBO initiative. Please check your eligibility here:

<https://www.embopress.org/page/journal/14602075/authorguide#chargesguide>

If you have any questions, please do not hesitate to contact the Editorial Office. Thank you again for this contribution to The EMBO Journal and congratulations on a successful publication!

Best wishes,

leva

leva Gailite, PhD
Senior Scientific Editor
The EMBO Journal
Meyerhofstrasse 1
D-69117 Heidelberg
Tel: +4962218891309
i.gailite@embojournal.org
